# ASTRAEA: A TOKEN-WISE ACCELERATION FRAMEWORK FOR VIDEO DIFFUSION TRANSFORMERS

**Haosong Liu**[1*]    **Yuge Cheng**[1*]    **Wenxuan Miao**[1]    **Zihan Liu**[1,2]    **Aiyue Chen**[3]    **Jing Lin**[3]
**Yiwu Yao**[3]    **Chen Chen**[1]    **Jingwen Leng**[1,2]    **Minyi Guo**[1,2]    **Yu Feng**[1,2,#]
[1]Shanghai Jiao Tong University    [2]Shanghai Qizhi Institute    [3]Huawei Technologies Co.,Ltd
[*]Equal Contribution    [#]Corresponding Author
{2436824987, chengyuge, miaowenxuan, altair.liu, chen-chen}@sjtu.edu.cn
{leng-jw, guo-my}@cs.sjtu.edu.cn
{chenaiyue, linjing28, yaoyiwu}@huawei.com
y-feng@sjtu.edu.cn
Project site: https://astraea-project.github.io/ASTRAEA/

## ABSTRACT

Video diffusion transformers (vDiTs) have made tremendous progress in text-to-video generation, but their high compute demands pose a major challenge for practical deployment. While studies propose acceleration methods to reduce workload at various granularities, they often rely on heuristics, limiting their applicability.

We introduce ASTRAEA, a framework that searches for near-optimal configurations for vDiT-based video generation under a performance target. At its core, ASTRAEA proposes a lightweight token selection mechanism and a memory-efficient, GPU-friendly sparse attention strategy, enabling linear savings on execution time with minimal impact on generation quality. Meanwhile, to determine optimal token reduction for different timesteps, we further design a search framework that leverages a classic evolutionary algorithm to automatically determine the distribution of the token budget effectively. Together, ASTRAEA achieves up to $2.4\times$ inference speedup on a single GPU with great scalability (up to $13.2\times$ speedup on 8 GPUs) while achieving up to over 10 dB video quality compared to the state-of-the-art methods ($<0.5\%$ loss on VBench compared to baselines).

## 1    INTRODUCTION

Visual imagination has always been at the core of humanity's nature for creativity. After the release of Sora by OpenAI in early (OpenAI, 2024), there are numerous video generative frameworks from text input, including Kuaishou's Kling (Kuaishou, 2024), Google's Veo (Google, 2024), and Tencent's HunyuanVideo (Tencent, 2024). Despite the abundance of frameworks, video diffusion transformers (vDiTs) remain the core of these frameworks, widely regarded as the most effective paradigm for high-fidelity video generation. However, its computational demand poses a challenge for any industrial-level deployment. For instance, a 4-second $720\times1280$ video clip using Hunyuan-Video (Kong et al., 2024) takes over 0.5 hours on a single Nvidia H100 GPU. This high computational cost comes from the *extensive denoising steps* and the *long token sequence*. For example, a vDiT model often requires 50-100 denoising steps, with each step involving millions of tokens.

To address the inference inefficiency, various acceleration methods have been proposed to reduce computations at different granularities, such as denoising step reduction (Li et al., 2023; Yin et al., 2024; Gu et al., 2023), block caching (Zhao et al., 2024; Chen et al., 2024; Kahatapitiya et al., 2024; Fang et al., 2023), token selection (Zou et al., 2024b), etc. However, these methods often require iteratively fine-tuning hyperparameters, i.e., which steps or blocks to skip, with a human in the loop to achieve a target performance in industrial-level deployments. Fundamentally, previous approaches propose various acceleration heuristics, without addressing a key question: *given a performance target, which tokens are worth computing at each denoising step to achieve optimal accuracy?*

To answer this question, we propose ASTRAEA, a GPU-friendly acceleration framework that operates at the token level, the smallest primitive in vDiT models. Specifically, we propose a sparse

Fig. 1: Diffusion models work by reversing a diffusion process, where they iteratively predict and remove noise at each timestep to gradually reconstruct the original images or videos.

diffusion inference with a lightweight *token selection mechanism* and *GPU-efficient sparse attention* to accelerate each denoising step by dynamically selecting important tokens. Meanwhile, to determine optimal token budget allocation, a search framework is proposed to determine the number of tokens that should be assigned in each denoising step to achieve the target performance.

**Algorithm.** One key downside of prior work (Zou et al., 2024b; Zhao et al., 2024) is the extensive GPU memory usage. Unlike prior studies, which require storing the entire attention map, our mechanism introduces negligible memory overhead by only caching previous tokens. Thus, the memory requirement of our selection metric scales linearly with the token length to avoid a memory explosion. In addition, we purposely design our sparse attention to be natively parallelizable; thus, it can be integrated with existing acceleration techniques, such as FlashAttention (Dao, 2023). On existing GPUs, our sparse attention achieves linear speedup with the number of reduced input tokens.

**Search Framework.** Although our algorithm can achieve linear acceleration for each denoising timestep, it is still unknown how many tokens should be selected for individual timesteps. While numerous search techniques exist in the literature, such as neural architecture search (White et al., 2023; Elsken et al., 2019) and network pruning (Hoefler et al., 2021; Cheng et al., 2024), none can be applied to vDiTs due to their substantial search times. Here, we propose a search framework based on the classic evolutionary search algorithms (Whitley et al., 1996; Ashlock, 2006). Our approach guarantees to achieve the target performance while achieving the minimal accuracy loss.

ASTRAEA achieves up to $2.4\times$ speedup with up to $>10$ dB better quality in PSNR compared to the state-of-the-art algorithms, and achieves $<0.5\%$ loss on VBench. ASTRAEA also shows great scalability and achieves $13.2\times$ speedup across 8 GPUs. Our contributions are summarized as follows:

- We propose a lightweight token selection mechanism to reduce the computation workload of each denoising step with negligible latency and memory overhead.
- We design a sparse attention computation method that achieves linear speedup on existing GPUs, as the number of selected tokens decreases.
- We introduce a search framework that meets the target performance while achieving the minimal accuracy loss against prior studies.

## 2 RELATED WORK

Diffusion models have been widely used in video generation. It learns to generate data from Gaussian noise through a reverse Markov process (Fig. 1). The input of this diffusion process is a randomly generated Gaussian noise, $x'_T$. The diffusion model recovers the original data by progressively predicting and removing noise, $z'_t$, from $x'_t$ at each timestep $t$. The hope is that, through this process, the prediction of the diffusion model, $x'_0$, can be close to the original data, $x_0$. Mathematically, the denoising step can be expressed as,

$$x'_{t-1} = \alpha_t(x'_t - \beta_t z'_t) + \sigma_t n'_t, \ z'_t = \Phi(x'_t, t), \tag{1}$$

where $\Phi$ is the prediction function that predicts the noise $z'_t$. Both $\alpha_t$ and $\beta_t$ are hyper-parameters. $\sigma_t n'_t$ is the renoising term to add randomness to the denoising step.

Normally, diffusion models often require hundreds or even thousands of steps to denoise images or videos (Yang et al., 2022). The common practice to reduce the diffusion workload is to encode the initial noising inputs into the latent space, then decode the tokens at the end (Fig. 1).

Prior methods to enhance the efficiency of diffusion models fall into two categories: *step reduction*, which reduces the number of denoising steps, and *block caching*, which seeks to minimize the computational demands within each denoising step. The following paragraphs provide an overview of these two categories of acceleration techniques separately.

**Step Reduction.** Overall, step reduction methods can be classified into two types: one requires retraining, and the other is training-free. The retraining methods (Yin et al., 2024; Salimans & Ho, 2022; Gu et al., 2023; Habibian et al., 2024; Kim et al., 2024), such as distillation, can reduce the number of denoising steps to as few as one. However, the major downside of these methods is that they require as much training time as the original model training. For this reason, current distillation methods are primarily applied to image generation rather than video generation (Salimans & Ho, 2022; Gu et al., 2023; Yin et al., 2024; Habibian et al., 2024).

On the other hand, training-free methods do not require re-training and are generally less effective. The intuition of these training-free methods is to leverage the insignificance of predicted noises between steps, allowing diffusion models to skip these less critical steps. For instance, PAB (Zhao et al., 2024) periodically skips two out of every three intermediate steps to reduce the overall computation. AutoDiffusion (Li et al., 2023) applies heuristic search to iteratively develop better step-sampling strategies. FasterCache (Lv et al., 2024) and Gradient-Optimized Cache (Qiu et al., 2025) both propose some caching mechanisms to accelerate video diffusion. Fast Video Generation (Zhang et al., 2025c) introduces a tile-based attention pattern that accelerates video diffusion without requiring model retraining. AdaDiff (Zhang et al., 2023) and similar work (Fang et al., 2023) dynamically select steps to skip during inference.

**Block Caching.** The execution time of diffusion transformers is dominated by either 2D or 3D attention (Lin et al., 2024; Zheng et al., 2024). To reduce the computation in self-attention, existing methods exploit the similarity of intermediate results across denoising steps and cache the intermediate results to avoid extra computations (Zhao et al., 2024; Chen et al., 2024; Kahatapitiya et al., 2024; Ma et al., 2024; Wimbauer et al., 2024; Liu et al., 2024a;b).

Primary caching methods operate at the block level. They reuse the intermediate result of a block from the previous denoising step and skip the corresponding block computation in the current denoising step entirely (Zhao et al., 2024; Kahatapitiya et al., 2024; Liu et al., 2024b; Chen et al., 2024; Liu et al., 2024a; Zhang et al., 2025b; Ma et al., 2024; Wimbauer et al., 2024; Zhang et al., 2025a). The key difference among block-wise methods lies in the strategies they apply to reuse the intermediate results. For instance, PAB (Zhao et al., 2024) applies a fixed step-skipping scheme throughout the entire inference process, whereas $\Delta$-DiT (Chen et al., 2024) and AdaptiveCache (Kahatapitiya et al., 2024) adopt an adaptive scheme to tailor the step reuse for each input. SmoothCache (Liu et al., 2024b) and BlockDance (Zhang et al., 2025b) exploit different features in DiTs to enable continuous reuse of transformer blocks. Nevertheless, the underlying concept remains unchanged.

Recently, a few studies (Zou et al., 2024b; Bolya & Hoffman, 2023; Zou et al., 2025) have started to explore reuse at a finer granularity, the token level. For instance, ToCa (Zou et al., 2024b) proposes a composed metric that identifies the unimportant tokens during the inference and uses the previously cached results for attention computation. However, its token selection process introduces non-trivial compute and memory overheads. Neither is affordable on modern GPUs. Thus, token-wise methods are still not fully exploited and require careful algorithmic design to enable practical usage.

## 3 METHODOLOGY

This section describes our framework, ASTRAEA. The goal of ASTRAEA is to determine which tokens are worth computing throughout the diffusion process. Sec. 3.2 first describes our selection method that determines which specific tokens should be computed at each timestep under a token budget. Sec. 3.3 then describes a search algorithm that determines the token budget at each timestep.

### 3.1 PRELIMINARY

**Self-Attention.** We first introduce one of the key operations in vDiTs: self-attention. The input to the self-attention, $X_{\text{in}}$, is a sequence of tokens with a shape of $\langle N, d \rangle$. $N$ is the number of tokens and $d$ is the token channel dimension. The computation of self-attention can be expressed as,

$$\text{Attention}(Q, K, V) = \text{Softmax}(\frac{QK^T}{\sqrt{d_k}})V, \tag{2}$$

where query $Q$, key $K$, and value $V$ are generated by performing three independent linear projections on input tokens, $X_{\text{in}}$. These three values have the same dimensions as $X_{\text{in}}$. The product $QK^T$

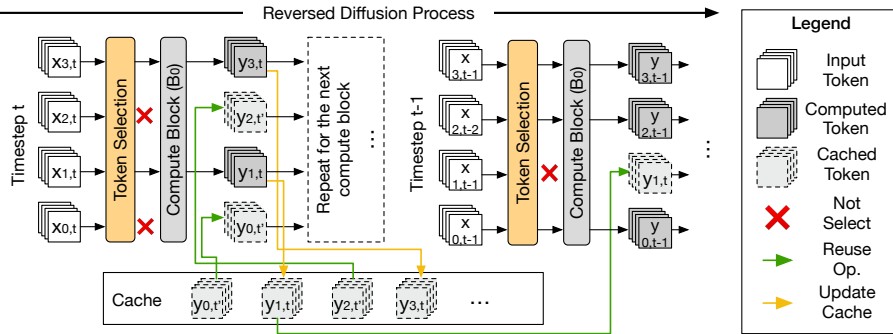

Fig. 2: The general diffusion process with token caching. For each compute block, a selection module determines which tokens should be computed. Only the selected tokens perform computation. The unselected tokens skip the compute block and query the cache for their results.

is commonly referred to as the attention map, $A$, with a shape of $\langle N, N \rangle$. The attention map is then divided by $\sqrt{d_k}$ where $d_k$ is the channel dimension of $K$. Finally, each row of the attention map, $A_i$, is normalized by the softmax function, where

$$\text{Softmax}(A_{i,j}) = \frac{e^{A_{i,j}}}{\sum_{k=1}^{N} e^{A_{i,k}}}, \tag{3}$$

before performing a dot product with $V$. $i$ and $j$ are the row and column indices of the attention map, $A$. Note that, $\sum_{k=1}^{N} e^{A_{i,k}}$ is often called the Log-Sum-Exp (LSE) score.

## 3.2 TOKEN SELECTION

**Execution Flow.** Fig. 2 illustrates how our token selection integrates into the general diffusion process. For instance, at timestep $t$, there are four input tokens, $\langle x_{0,t}, ..., x_{3,t} \rangle$. Before these tokens pass through a compute block (typically a stack of a self-attention layer, a cross-attention layer, and a multilayer perceptron in vDiTs), the token selection module first determines which tokens should be computed. This module computes the importance of each token and selects the top significant tokens, under a pre-defined computation budget, $\theta^*$.

The selected tokens are then processed through the compute block, and their outputs are used to update the cache. In contrast, unselected tokens would directly query their outputs from the cache, which stores results from the same compute block at earlier timesteps. The final output token sequence is a combination of both computed and cached tokens, e.g., $\langle y_{0,t'}, y_{1,t}, y_{2,t'}, y_{3,t} \rangle$ in Fig. 2. The same process is applied to all compute blocks in the subsequent process.

**Selection Metric.** Given the execution flow above, we next describe our token selection mechanism. The goal of token selection is to skip unimportant tokens while retaining the same generation quality. Despite studies (Zou et al., 2024b; Bolya et al., 2022; Bolya & Hoffman, 2023) proposing various token selection metrics, they *either incur high computational and memory overhead (Zou et al., 2024b) or are specifically tailored to particular tasks (Bolya et al., 2022; Bolya & Hoffman, 2023)*. Thus, we propose a general token selection metric, $S_{\text{token}}$, that applies to all vDiTs. Tbl. 1 shows that $S_{\text{token}}$ achieves superior performance compared to prior metrics.

Mathematically, $S_{\text{token}}$ has two components,

$$S_{\text{token}} = w_\alpha S_{\text{sig}} + w_\beta S_{\text{penalty}}, \tag{4}$$

where $S_{\text{sig}}$ stands for the significance of this token and $S_{\text{penalty}}$ represents the penalty for repeatedly choosing the same token across multiple timesteps. Both $w_\alpha$ and $w_\beta$ are the hyperparameters that are used to weigh the contributions of $S_{\text{sig}}$ and $S_{\text{penalty}}$. $S_{\text{sig}}$ is expressed as,

$$S_{\text{sig}} = S_{\text{LSE, t-1}} \Delta_{\text{token, t}}, \tag{5}$$

where $S_{\text{LSE, t-1}}$ is the LSE score of each token computed in the softmax function in the previous timestep, $t-1$. Here, we use the LSE score in the previous timestep because LSE scores do not vary across timesteps. To verify this point, we calculate the cosine similarity of the LSE of the attention for each block in adjacent timesteps during the inference process of the Wan 2.1 (Wang et al., 2025). The average cosine similarity of adjacent LSE is 99.1% with a standard error of 0.00736%.

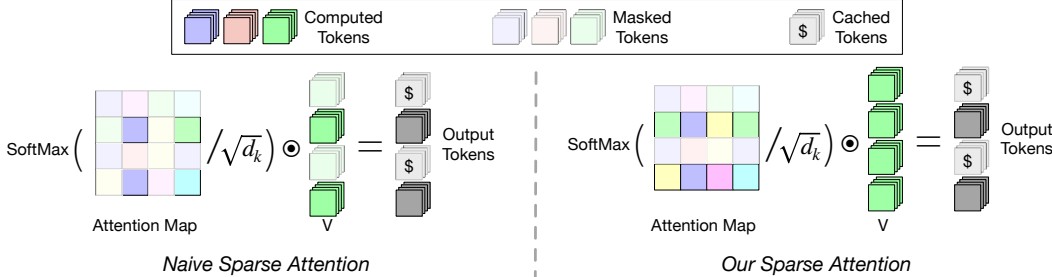

Fig. 3: An illustration of our sparse attention computation with selected tokens. While naive sparse attention reduces overall computation quadratically by computing the selected tokens on $Q$, $K$, and $V$, it suffers from computational inaccuracies and requires excessive memory usage. Our sparse attention only computes the selected tokens on $Q$ and keeps all tokens for $K$ and $V$. The pseudocode of our sparse attention is shown in Algo. 1 in the appendix.

Also, $S_{\text{LSE, t-1}}$ is included in $S_{\text{sig}}$ because its value is proportional to the attention score in self-attention, which reflects the token importance. Meanwhile, $S_{\text{LSE, t-1}}$ is the byproduct of softmax and incurs no additional computational overhead.

$\Delta_{\text{token, t}}$ is the value difference of individual input tokens across two adjacent computed timesteps. Here, a timestep is considered computed if the token is evaluated at that timestep rather than reused from cache. Note that, we calculate $\Delta_{\text{token, t}}$, the difference of input tokens, so that we can select the important tokens without performing the attention computation. $S_{\text{penalty}}$ is expressed as,

$$S_{\text{penalty}} = e^{n_i}, \tag{6}$$

where $n_i$ is the number of times the $i$th token has not been selected consecutively. This penalty term is inspired by ToCa (Zou et al., 2024b), and we claim no contribution.

**Sparse Attention.** Next, we describe how to perform vDiT inference with the selected tokens. There are three main computation operations in vDiT: self-attention, cross-attention, and multilayer perceptron (MLP) (Zheng et al., 2024; Lin et al., 2024). Both cross-attention and MLP operate on individual tokens; thus, we can directly skip the unselected tokens to reduce the computation.

However, naively performing sparse self-attention with selected tokens, e.g., in ToCa (Zou et al., 2024b), would alter the semantics of self-attention, as shown in Fig. 3. Only computing the unmasked positions in the attention map would lead to two issues: incorrect results and substantial memory overhead. This is because self-attention requires calculating the LSE score for each row in the attention map. Just computing the unmasked positions is semantically incorrect. Even reusing the results of the same attention in a previous denoising timestep would lead to accuracy loss. Meanwhile, it requires caching the entire attention map, which introduces high memory overhead. Tbl. 1 shows that our technique achieves smaller memory overhead (roughly $2\times$ reduction) and much higher accuracy ($> 10$dB in PSNR) against the prior methods.

In contrast, we propose a seemingly "counterintuitive" sparse attention computation, as shown in Fig. 3, where we only selectively compute the queries $Q$ while computing all tokens for the keys $K$ and values $V$. Although this approach saves less computation than naive sparse attention, it offers the following advantages. First, by computing the entire row of elements in the attention map, we ensure the individual output token is computed correctly. Second, our sparse attention is natively GPU-parallelizable and can be integrated with existing attention acceleration techniques, such as FlashAttention (Dao et al., 2022). Third, our sparse attention does not require any additional GPU memory, except for the cached tokens, which are negligible compared to the all attention scores.

### 3.3 TOKEN-WISE SEARCH FRAMEWORK

**Problem Setup.** Sec. 3.2 explains how to select tokens of a compute block under a given token budget, $\theta^*$. The next question is what token budget should be allocated for each compute block. Ideally, to achieve the best performance, we should search for token budgets at the block level. However, searching at the block level would make the search space intractable. Thus, we fix the token budget at the timestep level and formulate the problem as follows: *Given a total number of selected tokens, how should the token budget be allocated for each timestep?*

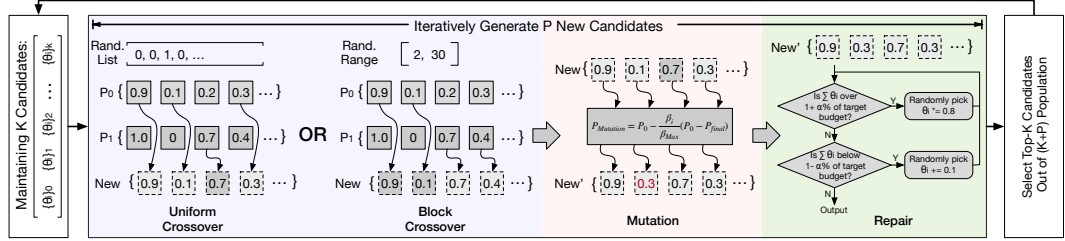

Fig. 4: An illustration of EA with its three key steps. Each new candidate would go through these three steps sequentially before being added to the existing population. For each candidate, we would randomly pick one out of two crossover methods. The detailed EA procedure is shown in Algo. 2.

**Search Space.** In our framework, the search space is $\Theta$, which can expressed as,

$$\Theta = \{\theta_i\}, i \in [1, 2, ..., T] \text{ and } \theta_i \in \{0, 10\%, 20\%, ..., 100\%\}, \tag{7}$$

where $\theta_i$ is the percentage of selected tokens at the denoising timestep $i$. $T$ is the maximal timestep.

**Algorithm.** We now introduce our search algorithm, which adopts the classic stochastic search framework, evolutionary algorithm (EA) (Whitley et al., 1996; Ashlock, 2006), to search for the optimal token allocation across denoising steps. EA simulates the process of natural selection, which evolves a population of candidates over generations using operations, e.g., selection, crossover, and mutation, to find a near-optimal solution. In EA, we start by spawning the initial generation of $K$ candidates. We set $K$ to be 50 in this paper. Each candidate has a list of selected token percentages, $\{\theta_i\}_k, k \in [0, K)$. Each $\theta_i$ stands for selecting $\theta_i$ percent of tokens at the $i$th timestep. All $\{\theta_i\}_k$ values together need to fit the total token budget constraint, $\Theta_\$$.

At each generation, the top-$K$ number of parents with smaller MSE losses are selected from the previous generation. We then generate $P$ number of new candidates from these top-$K$ parents ($P = 30$). For each new candidate, we randomly pick a parent pair from the top-$K$ parents. The selected parent pair then goes through three key steps: crossover, mutation, and repair, to generate those new candidates. These newly generated candidates are added to the current population. These new candidates are then evaluated based on the MSE between the original video output and the output generated using the selected tokens. Finally, the top-$K$ candidates with the lowest MSEs are selected for the next generation. After $G$ generations ($G = 30$), we will pick the best candidate as the solution for our token allocation. Next, we explain the three key steps in this process, as shown in Fig. 4.

*Crossover.* This step aims to generate a new candidate by combining two randomly picked candidates from the previous population. We propose two crossover strategies: *uniform crossover* and *block crossover*. Given two parent candidates, $\{\theta_i\}_{p0}$ and $\{\theta_i\}_{p1}$, uniform crossover randomly selects each $\theta_i$ from either parent with equal probability to form a new candidate. In contrast, block crossover first randomly creates a contiguous subset of timesteps within the range $[0, T]$. The new candidate then inherits all $\theta_i$ values within this subset from one parent, and the remaining values from the other parent. In the crossover step, we randomly pick between uniform crossover and block crossover.

*Mutation.* Once a new candidate $\{\theta_i\}_{\text{new}}$ is generated, the goal of mutation is to introduce a possibly better candidate by randomly mutating this candidate. We decide whether each timestep would be mutated based on the probability, $P_{\text{mutation}} = P_0 - \frac{\beta_i}{\beta_{\text{Max}}}(P_0 - P_{\text{final}})$, where $P_0$ and $P_{\text{final}}$ are the initial and the final probability of mutation, respectively. We find that gradually decreasing the mutation probability over generations leads to better convergence. $\beta_i$ is the $i$th evolution generation and $\beta_{\text{max}}$ is the maximal evolution generation. If a timestep were mutated, our algorithm would randomly change its $\theta_i$ from the valid value range, i.e., $\{0, 0.1, ..., 1.0\}$.

*Repair.* A new candidate $\{\theta_i\}_{\text{new}'}$ from the previous two steps might no longer satisfy the token budget constraint, $\Theta_\$$. This repair step would ensure that the total token budget falls within the acceptable range, $[0.9\Theta_\$, 1.1\Theta_\$]$. If the total budget exceeds the upper bound, we randomly decrease one or more $\theta_i$. Conversely, if the total budget is below the lower bound, we randomly increase one or more $\theta_i$ values until the constraint is met, as shown in Fig. 4.

**Generality.** Standard EA typically requires generating ground truth videos for multiple sample prompts, which introduces additional computational overhead. In our implementation, we simply select 4 prompts from different genres. While including more prompts could theoretically improve generalization, we observe that it leads to minimal improvement in search outcomes while sub-

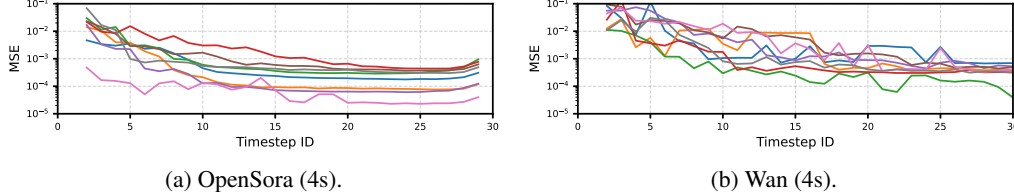

(a) OpenSora (4s).            (b) Wan (4s).

Fig. 5: Different prompts show a similar trend when removing one specific timestep out of the entire denoising process. The MSE is calculated against the original result without skipping timesteps.

stantially increasing the search cost. The reason is that different prompts often exhibit a similar robustness trend as shown in Fig. 5. Specifically, for each prompt, we skip one timestep during the denoising process and calculate the MSE loss against the ground truth. By sweeping the timesteps, we find that different prompts show a similar MSE trend on both OpenSora and Wan (Fig. 5).

**Why EA?** There are two main reasons why we choose EA over other search methods. First, EA keeps the original model intact. Search methods, such as NAS (Elsken et al., 2019) or network pruning (Cheng et al., 2024), would modify the network architecture or model weights. In contrast, EA can well fit in the token selection problem without altering the original vDiT models. Second, both NAS and Network Pruning are several orders of magnitude more computationally intensive than ours. For instance, NASNet (Zoph et al., 2018) requires approximately 2,000 GPU hours to complete its architecture search, and Diff-Pruning] (Fang et al., 2023) demanded up to 20% of the original model training time. In contrast, our work only requires, on average, 82 GPU hours. So we choose lightweight EA instead.

## 4    EVALUATION

### 4.1    EXPERIMENTAL SETUP

**Baselines.** We evaluate three widely used text-to-video generation frameworks: HunyuanVideo-T2V (Kong et al., 2024), Wan v2.1 1.3B (Wang et al., 2025) and OpenSora v1.2 (Zheng et al., 2024). For HunyuanVideo-T2V, we generate 5-second videos with a resolution of $544 \times 960$. For Wan and OpenSora, we test both short-sequence videos (2-second 480P) and long-sequence videos (4-second 480P). Our evaluation compares with five state-of-the-art training-free methods: $\Delta$-DIT (Chen et al., 2024), PAB (Zhao et al., 2024), TOCA (Zou et al., 2024b), SVG (Xi et al., 2025), and SVG2 (Yang et al., 2025).

**EA Configuration.** In our EA algorithm, we set the maximal generation to be 30, $K$ and $P$ are set to be 50 and 30 in each generation. To ensure diversity of candidates in the early stages and structural stability of good candidates in the later stages, we set $P_0$ and $P_{final}$ to be 0.1 and 0.01, respectively. Across all four evaluated models, the results converge after searching 30 generations.

**Metrics and Hardware.** Following prior works (Zhao et al., 2024; Zou et al., 2024b; Xi et al., 2025; Chen et al., 2024), we use the VBench score (Huang et al., 2024) as the video quality metric. During the experiments, we generate 5 videos for each of the 950 benchmark prompts using different random seeds. The generated videos are then evaluated across 16 aspects. We report the average value of the aspects. In addition, we compare the generated videos by different acceleration methods against the original video results frame-by-frame on image quality, using PSNR, SSIM, and LPIPS. For performance, we report end-to-end generation latency, GPU memory consumption, and computational complexity (FLOPs). Our evaluation uses two GPUs as our hardware platforms: Nvidia A6000 with 48 GB of memory and Nvidia A100 with 80 GB of memory.

### 4.2    PERFORMANCE AND ACCURACY

**Video Quality.** Tbl. 1 presents the overall comparison of generation quality with different methods. Across all vDiT models, ASTRAEA achieves the highest speedup ($2.4\times$) while maintaining the best VBench score compared to other baselines. On the VBench metric, almost all ASTRAEA variants retain the accuracy loss within 0.5%. In contrast, the strongest baseline, TOCA$_{2,85\%}$, achieves only 79.2%, which is 1.0% lower than the original model's score on Wan (4s). Similarly, on OpenSora,

Table 1: Quantitative evaluation of our method against the state-of-the-arts (Chen et al., 2024; Zou et al., 2024b; Zhao et al., 2024; Xi et al., 2025; Yang et al., 2025) on three vDiT models: HunyuanVideo-T2V (Kong et al., 2024), Wan v2.1 1.3B (Wang et al., 2025), and OpenSora v1.2 (Zheng et al., 2024). ① and ② denote the **best** and second-best results among all methods, respectively. PAB: The subscript numbers represent the reuse strides for spatial, temporal, and cross-attention. ToCA: The subscript numbers denote the timestep reuse stride and MLP reuse ratio. SVG/SVG2: The percentage indicates the skipped attention computation. ASTRAEA: The percentage indicates the total token budget. HunyuanVideo cannot run on Nvidia A6000.

| Model | Metric / Method | Quality Metrics | | | | Performance Metrics | | | | | |
|---|---|---|---|---|---|---|---|---|---|---|---|
| | | VBench (%)↑ | PSNR (dB)↑ | SSIM↑ | LPIPS↓ | FLOPs ($10^{15}$)↓ | $L_{A100}$ (sec.)↓ | Speedup (A100) | $L_{A6000}$ (sec.)↓ | Speedup (A6000) | Mem. (GB)↓ |
| HunyuanVideo (5s) | Original | 80.28 | - | - | - | 217.27 | 1226.99 | 1.00 | - | - | 45.81 |
| | $\Delta$-DiT (Chen et al., 2024) | 79.43 | 26.09 | 0.862 | 0.111 | 145.09 | 996.44 | 1.23 | - | - | 46.59 |
| | PAB$_{2,6}$ (Zhao et al., 2024) | 79.64 | ②29.91 | 0.906 | 0.082 | 176.40 | 1048.82 | 1.17 | - | - | 66.17 |
| | PAB$_{5,9}$ (Zhao et al., 2024) | 78.90 | 26.07 | 0.778 | 0.084 | 145.55 | 1009.10 | 1.22 | - | - | 66.17 |
| | ToCA (Zou et al., 2024b) | - | - | - | - | - | - | - | - | - | OOM |
| | SVG$_{70\%}$ (Xi et al., 2025) | 79.97 | 25.78 | 0.843 | 0.153 | ②121.51 | 726.03 | 1.69 | - | - | 50.87 |
| | SVG2$_{70\%}$ (Yang et al., 2025) | ①80.80 | 21.52 | 0.747 | 0.259 | ②121.51 | 804.23 | 1.53 | - | - | 55.61 |
| | ASTRAEA$_{40\%}$ | 79.79 | 27.61 | 0.895 | 0.076 | ①113.27 | ①514.84 | ①2.38 | - | - | 69.01 |
| | ASTRAEA$_{50\%}$ | 80.20 | 28.71 | ②0.913 | ②0.058 | 130.10 | ②636.35 | ②1.93 | - | - | 69.01 |
| | ASTRAEA$_{70\%}$ | ②80.43 | ①33.62 | ①0.953 | ①0.026 | 163.77 | 881.55 | 1.39 | - | - | 69.01 |
| Wan (2s) | Original | 81.46 | - | - | - | 7.29 | 68.48 | 1.00 | 108.30 | 1.00 | 7.8 |
| | $\Delta$-DiT (Chen et al., 2024) | 78.37 | 15.13 | 0.499 | 0.408 | 5.87 | 60.52 | 1.13 | 87.81 | 1.23 | 7.81 |
| | PAB$_{2,6}$ (Zhao et al., 2024) | 80.05 | 18.02 | 0.667 | 0.246 | 4.67 | 50.36 | 1.36 | 79.84 | 1.35 | 11.59 |
| | PAB$_{5,9}$ (Zhao et al., 2024) | 78.61 | 17.60 | 0.638 | 0.290 | ②3.34 | 40.31 | 1.70 | 66.47 | 1.62 | 11.59 |
| | ToCA$_{2,80\%}$ (Zou et al., 2024b) | 81.06 | 18.01 | 0.651 | 0.254 | 4.14 | 44.43 | 1.54 | 75.02 | 1.44 | 17.66 |
| | ToCA$_{2,85\%}$ (Zou et al., 2024b) | 80.89 | 18.02 | 0.653 | 0.252 | 4.07 | 43.86 | 1.56 | 71.28 | 1.52 | 17.66 |
| | SVG$_{70\%}$ (Xi et al., 2025) | 77.51 | 18.87 | 0.695 | 0.237 | 4.11 | 56.58 | 1.21 | 100.08 | 1.08 | 21.99 |
| | SVG2$_{70\%}$ (Yang et al., 2025) | 79.48 | 22.23 | 0.787 | 0.127 | 4.11 | 60.09 | 1.14 | 95.84 | 1.13 | 20.70 |
| | ASTRAEA$_{40\%}$ | 80.82 | 23.77 | 0.826 | 0.144 | ①3.05 | ①30.23 | ②2.27 | ①46.05 | ②2.35 | 9.04 |
| | ASTRAEA$_{50\%}$ | ②81.11 | ②25.67 | ②0.884 | ②0.071 | 3.85 | ②37.29 | ②2.01 | ②56.77 | ①1.91 | 9.04 |
| | ASTRAEA$_{70\%}$ | ①81.28 | ①30.83 | ①0.948 | ①0.026 | 5.38 | 44.71 | 1.53 | 77.91 | 1.39 | 9.04 |
| Wan (4s) | Original | 80.28 | - | - | - | 19.87 | 155.01 | 1.00 | 253.62 | 1.00 | 8.97 |
| | $\Delta$-DiT (Chen et al., 2024) | 76.81 | 16.14 | 0.602 | 0.376 | 15.96 | 135.96 | 1.14 | 205.17 | 1.24 | 8.97 |
| | PAB$_{2,6}$ (Zhao et al., 2024) | 78.76 | 19.95 | 0.761 | 0.194 | 12.79 | 113.41 | 1.37 | 183.37 | 1.38 | 15.96 |
| | PAB$_{5,9}$ (Zhao et al., 2024) | 77.71 | 19.44 | 0.739 | 0.234 | ②8.99 | 90.72 | 1.71 | 148.58 | 1.71 | 15.96 |
| | ToCA$_{2,80\%}$ (Zou et al., 2024b) | 79.01 | 18.10 | 0.689 | 0.269 | 11.04 | 96.84 | 1.60 | 154.83 | 1.64 | 38.40 |
| | ToCA$_{2,85\%}$ (Zou et al., 2024b) | 79.28 | 18.13 | 0.694 | 0.264 | 10.92 | 95.07 | 1.63 | 152.34 | 1.66 | 38.40 |
| | SVG$_{70\%}$ (Xi et al., 2025) | 77.74 | 18.85 | 0.678 | 0.255 | 11.69 | 129.50 | 1.20 | 216.50 | 1.17 | 22.38 |
| | SVG2$_{70\%}$ (Yang et al., 2025) | 79.17 | 22.95 | 0.833 | 0.116 | 11.69 | 116.55 | 1.33 | 193.60 | 1.31 | 21.27 |
| | ASTRAEA$_{40\%}$ | 79.78 | 26.98 | 0.901 | 0.072 | ①8.20 | ①67.61 | ②2.29 | ①106.65 | ②2.38 | 11.71 |
| | ASTRAEA$_{50\%}$ | ②79.96 | ②28.12 | ②0.918 | ②0.053 | 10.32 | ②83.34 | ②1.86 | ②132.62 | ①1.91 | 11.71 |
| | ASTRAEA$_{70\%}$ | ①80.18 | ①33.00 | ①0.958 | ①0.021 | 14.42 | 114.20 | 1.36 | 184.89 | 1.37 | 11.71 |
| OpenSora (2s) | Original | 78.14 | - | - | - | 3.29 | 54.09 | 1.00 | 78.10 | 1.00 | 14.89 |
| | $\Delta$-DiT (Chen et al., 2024) | ①78.09 | 29.09 | 0.906 | 0.066 | 2.84 | 52.83 | 1.02 | 77.23 | 1.01 | 23.78 |
| | PAB$_{246}$ (Zhao et al., 2024) | 77.50 | 26.78 | 0.884 | 0.089 | 2.91 | 44.09 | 1.23 | 59.87 | 1.31 | 27.20 |
| | PAB$_{579}$ (Zhao et al., 2024) | 75.52 | 22.60 | 0.800 | 0.191 | 2.53 | 37.68 | 1.44 | 55.75 | 1.40 | 27.20 |
| | ToCA$_{3,80\%}$ (Zou et al., 2024b) | 77.13 | 20.28 | 0.766 | 0.209 | 1.89 | 32.04 | 1.69 | 53.61 | 1.45 | 41.27 |
| | ToCA$_{3,85\%}$ (Zou et al., 2024b) | 76.89 | 20.02 | 0.760 | 0.216 | 1.84 | 31.74 | 1.70 | 52.89 | 1.48 | 41.27 |
| | ASTRAEA$_{40\%}$ | 76.95 | 27.23 | 0.875 | 0.095 | ①1.50 | ①22.97 | ②2.35 | ①33.67 | ②2.32 | 20.08 |
| | ASTRAEA$_{50\%}$ | 77.45 | ②29.52 | ②0.908 | ②0.067 | ②1.82 | ②28.36 | ②1.91 | ②41.13 | ②1.82 | 20.08 |
| | ASTRAEA$_{70\%}$ | ②78.08 | ①31.78 | ①0.932 | ①0.039 | 2.48 | 37.15 | 1.46 | 54.54 | 1.43 | 20.08 |
| OpenSora (4s) | Original | 79.00 | - | - | - | 6.59 | 109.15 | 1.00 | 173.07 | 1.00 | 16.96 |
| | $\Delta$-DiT (Chen et al., 2024) | ②78.46 | 28.15 | 0.886 | 0.084 | 5.68 | 108.93 | 1.00 | 171.84 | 1.01 | 25.83 |
| | PAB$_{246}$ (Zhao et al., 2024) | 78.40 | ②28.65 | ②0.896 | ②0.081 | 5.82 | 76.48 | 1.43 | 139.52 | 1.24 | 41.55 |
| | PAB$_{579}$ (Zhao et al., 2024) | 76.63 | 23.36 | 0.804 | 0.192 | 5.10 | 70.71 | 1.54 | 129.52 | 1.34 | 41.55 |
| | ToCA$_{3,80\%}$ (Zou et al., 2024b) | 77.69 | 21.02 | 0.773 | 0.212 | 3.79 | 65.48 | 1.67 | OOM | OOM | 61.17 |
| | ToCA$_{3,85\%}$ (Zou et al., 2024b) | 77.68 | 20.72 | 0.767 | 0.219 | ②3.56 | 64.46 | 1.69 | OOM | OOM | 61.17 |
| | ASTRAEA$_{40\%}$ | 76.62 | 25.65 | 0.841 | 0.145 | ①3.00 | ①47.30 | ②2.31 | ①74.63 | ②2.32 | 27.98 |
| | ASTRAEA$_{50\%}$ | 78.07 | 28.51 | 0.891 | 0.086 | 3.65 | ②58.62 | ②1.86 | ②92.54 | ②1.87 | 27.98 |
| | ASTRAEA$_{70\%}$ | ①78.65 | ①30.92 | ①0.920 | ①0.056 | 4.97 | 76.13 | 1.43 | 121.57 | 1.42 | 27.98 |

Fig. 6: VBench metrics, speedup, and memory consumption of ASTRAEA against other methods.

we can achieve the best accuracy while achieving the highest speedup. Although $\Delta$-DiT achieves the best VBench score on OpenSora (2s), $\Delta$-DiT achieves no speedup. In contrast, ASTRAEA$_{70\%}$ is almost the best on all quality metrics with much higher speedup. ASTRAEA$_{50\%}$ achieves the third

Table 2: The ablation study on Wan (4s).

| Method | Quality Metrics | | | | Performance Metrics | | | | | |
|---|---|---|---|---|---|---|---|---|---|---|
| | VBench (%)↑ | PSNR (dB)↑ | SSIM↑ | LPIPS↓ | FLOPs $(10^{15})$↓ | $L_{A100}$ (sec.)↓ | Speedup (A100) | $L_{A6000}$ (sec.)↓ | Speedup (A6000) | Mem. (GB)↓ |
| Original | 80.28 | - | - | - | 16.54 | 155.01 | - | 253.62 | - | 8.97 |
| SELECTQ&K | 79.01 | 18.10 | 0.689 | 0.269 | 11.04 | 96.84 | 1.60 | 154.83 | 1.64 | 38.40 |
| TIMSTEP-LEVEL | 79.50 | 22.71 | 0.802 | 0.169 | 10.32 | 78.51 | 1.97 | 127.72 | 1.99 | 8.97 |
| FIXED-TOKEN | 77.92 | 19.75 | 0.753 | 0.214 | 10.32 | 83.20 | 1.86 | 132.19 | 1.91 | 11.71 |
| ASTRAEA | 79.96 | 28.12 | 0.918 | 0.053 | 10.32 | 83.34 | 1.86 | 132.62 | 1.91 | 11.71 |

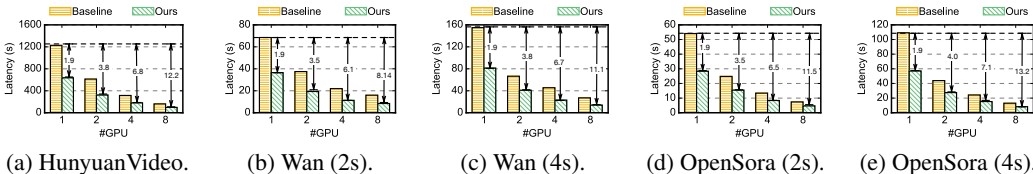

Fig. 7: VBench metrics, speedup, and memory consumption of ASTRAEA against other methods.

best on quality metrics while achieving higher speedup with a large margin. Sec. C .6 shows detailed VBench scores on five vDiT models. Across VBench scores, our variants are closely matched with the original baselines. The qualitative comparison of ASTRAEA against other methods is shown in the Appendix. Qualitatively, ASTRAEA achieves the best consistency against the original models.

(a) HunyuanVideo.  (b) Wan (2s).  (c) Wan (4s).  (d) OpenSora (2s).  (e) OpenSora (4s).

Fig. 8: The speedup of ASTRAEA against the baseline models across various numbers of GPUs.

**Image consistency.** In addition to VBench scores, we also perform frame-to-frame comparisons against the outputs from the original models to assess image consistency. Our results show that AS-TRAEA consistently preserves higher image consistency across all evaluated models. In particular, ASTRAEA outperforms all methods by a significant margin on Wan (4s) across all image quality metrics. On Wan, ASTRAEA $_{70\%}$ and ASTRAEA $_{50\%}$ outperform the strongest baseline by 10 dB.

**Performance.** Tbl. 1 also shows the performance comparison across various models. ASTRAEA consistently outperforms all baselines in both inference speed and GPU memory. Across models, ASTRAEA $_{40\%}$ delivers the highest speedup 2.4× on both A100 and A6000 with the lowest memory usage. We configure all baselines to achieve their best quality; however, none of them can achieve over 2× speedup. In contrast, ASTRAEA can be easily configured to achieve any target speedup (see sensitivity study in Sec. 4 .4) with the minimal quality tradeoff.

**Execution Breakdown.** Fig. 7 shows the execution breakdown of three vDiT models: three vDiT models: HunyuanVideo-T2V (Kong et al., 2024), Wan v2.1 1.3B (Wang et al., 2025), and OpenSora v1.2 (Zheng et al., 2024). Here, Wan and OpenSora are all 4-second videos. From Fig. 7, the latencies of all original compute blocks are proportionally decreased via our token selection technique. In addition, our token selection only takes 2.3% of the total execution time. Results indicate that our selection mechanism has low overhead.

**Scalability.** ASTRAEA shows strong performance scalability across various vDiT models. As shown in Fig. 8, our method demonstrates sublinear speedups as the number of GPUs increases across four different models. Specifically, our ASTRAEA $_{50\%}$ can achieve 13.2× speedup on Open-Sora with 8 GPUs. Overall, ASTRAEA can achieve over 10× speedup with 8 GPUs across all models. This shows the high parallelizability of our sparse attention described in Sec. 3 .2.

## 4 .3 ABLATION STUDY

In the ablation study, we compare three different variants: SELECTQ&K, TIMSTEP-LEVEL and FIXED-TOKEN. SELECTQ&K sparsely select both Q and K for every block (similar to naive sparse

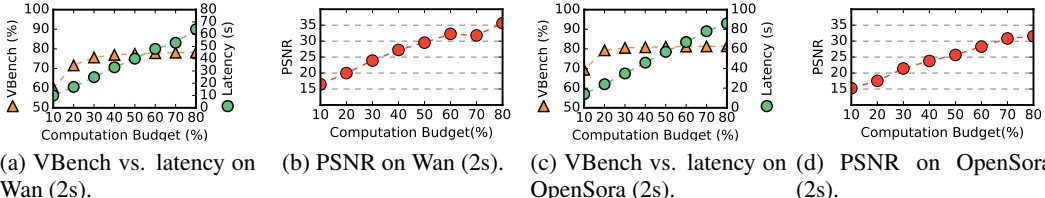

(a) VBench vs. latency on Wan (2s).    (b) PSNR on Wan (2s).    (c) VBench vs. latency on OpenSora (2s).    (d) PSNR on OpenSora (2s).

Fig. 9: Sensitivity of quality metrics and performance to computational budget percentage.

attention). TIMSTEP-LEVEL only selects timesteps. Each timestep either computes all tokens or skips computation entirely. FIXED-TOKEN selects tokens at the granularity of timesteps instead of blocks. All selected tokens within a timestep are computed, while unselected ones are skipped.

Our experiments show that all three variants achieve much lower VBench scores compared to our method. Specifically, FIXED-TOKEN drops the VBench score significantly ($>2.0\%$). This shows that the important tokens vary across compute blocks. On the other hand, TIMSTEP-LEVEL drops the VBench scores modestly, while achieving a slightly higher speedup compared to our method under the same token budget. This suggests that selecting at the timestep level may be a viable approach when trading off accuracy for higher performance. However, TIMSTEP-LEVEL suffers from noticeably lower image consistency compared to ASTRAEA.

### 4.4 SENSITIVITY STUDY

Fig. 9 shows the sensitivity of the computation budget (expressed as a percentage) to both the overall VBench score and execution latency on Wan (2s) and OpenSora (2s). On both models, we observe that the VBench score degrades rapidly when the computation budget drops below around 30%. In contrast, execution latency increases linearly with the computation budget. These results suggest that selecting a computation budget in the range between 30% and 50% offers a favorable trade-off.

Fig. 10 shows the sensitivity of the hyperparameters in Eqn. 4 to PSNR on Wan (4s). Recall, $w_\alpha$ is the hyperparameter for the significance term, $S_{\text{sig}}$, and $w_\beta$ is the hyperparameter for the penalty term, $S_{\text{penalty}}$, for non-selected tokens. In this sensitivity study, we fix $w_\alpha = 1$ and only vary $w_\beta$. The overall results show that ASTRAEA achieves the best quality when $w_\alpha$ and $w_\beta$ are 1 and 0.5, respectively. However, the overall variation is extremely small ($< 0.2$ PSNR), indicating that our method is robust to these hyperparameters. Also, we would like to emphasize that this penalty term is adapted directly from TOCA (Zou et al., 2024b), and our sensitivity trend matches the trend reported in TOCA (Zou et al., 2024b). We claim no contribution to this term.

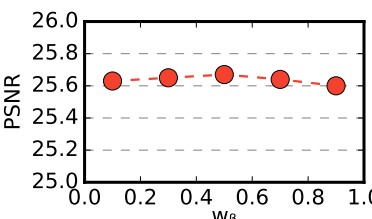

Fig. 10: The sensitivity of hyperparameters in Eqn. 4 on Wan (4s).

### 5 CONCLUSION

As vDiTs continue to drive breakthroughs in text-to-video generation, their deployment remains limited by computational demands. This work presents ASTRAEA, a framework that systematically accelerates vDiT inference through fine-grained token-level selection. By combining our three optimizations in Sec. 3, ASTRAEA dynamically determines the optimal token selection at each denoising step. We demonstrate that our method not only delivers almost linear speedup in inference latency under certain performance target but also preserves the highest generation quality.

### ACKNOWLEDGMENT

This work was supported by the National Natural Science Foundation of China (NSFC) Grants (62532006 and 62402312), Shanghai Qi Zhi Institute Innovation Program (SQZ202316), and Fundamental and Interdisciplinary Disciplines Breakthrough Plan of the Ministry of Education of China (JYB2025XDXM113).

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

---

**Algorithm 1** Efficient Attention via Temporal Token Selection.

---

**Require:** $X_{\text{in, t-1}}, X_{\text{in, t}} \in \mathbb{R}^{N \times D}$      ▷ Token features at timestep $t-1$ and $t$
**Require:** $X_{\text{out, t-1}} \in \mathbb{R}^{N \times D}$      ▷ Cached output tokens at timestep $t-1$
**Require:** $\theta$      ▷ The token budget for this attention block
**Require:** $W_Q, W_K, W_V$      ▷ Projection matrices
**Ensure:** $X_{\text{out, t}} \in \mathbb{R}^{N \times D}$      ▷ Updated token features at $t$
1: $S_{\text{sig}} \leftarrow \text{Mean}((X_{\text{in, t}} - X_{\text{in, t-1}}), \dim = 1) \cdot S_{\text{LSE,t-1}}$      ▷ Per-token squared difference
2: $S_{\text{token}} \leftarrow w_\alpha \cdot S_{\text{sig}} + w_\beta \cdot S_{\text{penalty}}$      ▷ Calculate our proposed selection metric
3: $I_{\text{sel}}, I_{\text{remain}} \leftarrow \text{TopK}(S_{\text{token}}, \theta)$      ▷ Selected top-$\theta$ percentile of token indices

4: $K_t \leftarrow X_{\text{in, t}} \cdot W_K, V_t \leftarrow X_{\text{in, t}} \cdot W_V$      ▷ Project K and V
5: $Q_{\text{sel, t}} \leftarrow X_{\text{in, t}}[I_{\text{sel}}] \cdot W_Q$      ▷ Select important Query tokens and perform projection

6: $X_{\text{out, t}}[I_{\text{sel}}] \leftarrow \text{Softmax}(Q_{\text{sel,t}} \cdot K^\top / \sqrt{d})V$      ▷ Compute attention only for selected Q
7: $X_{\text{out, t}}[I_{\text{remain}}] \leftarrow X_{\text{out, t-1}}[I_{\text{remain}}]$      ▷ Unselected tokens reuse the previously cached results
8: **return** $X_{\text{out, t}}$

---

## A    CODE OF ETHICS

I acknowledge that all co-authors of this work have read and commit to adhering to the ICLR Code of Ethics.

## B    LLM USAGE

We did not use LLM throughout the entire submission.

## C    SUPPLEMENTARY

### C.1    TOKEN SELECTION WITH SELF-ATTENTION.

Algo. 1 illustrates how our lightweight token selection mechanism is integrated with the self-attention computation. Overall, our algorithm dynamically selects tokens required for computation based on their significance and imposes a penalty for consecutive non-selected tokens.

**Inputs.** Our self-attention algorithm requires two kinds of input tokens, $X_{t-1}$ and $X_t$, which are the input tokens of the given attention block at the timestep $t-1$ and $t$, respectively. Meanwhile, our algorithm requires a token budget, $\theta$, which is between 0 and 100%. $\theta$ stands for select $\theta$-percent of top important tokens. $X_{\text{out, t-1}}$ is the previously cached token results from the early timesteps.

**Output.** The output of our attention is the output token sequence, $X_{\text{out, t}}$.

**Procedure.** We now describe the detailed process of how our token selection mechanism is integrated with the self-attention. The overall procedure consists of six steps.

- The first step is to compute the significance, $S_{\text{sig}}$, of each token. $S_{\text{sig}}$ is the weighted absolute difference between the input tokens of timestep $t-1$ and $t$. The absolute difference is then weighted by $S_{\text{LSE, t-1}}$.

- Once $S_{\text{sig}}$ is computed, we then obtain our selection metric, $S_{\text{token}}$, which is the weighted sum between $S_{\text{sig}}$ and $S_{\text{penalty}}$. Here, $S_{\text{penalty}}$ is a penalty term that prevents one particular token from being selected repeatedly.

- The third step is to find the top-$\theta$ percentile of token indices. Here, $I_{\text{sel}}$ are the indices of the selected tokens, and $I_{\text{remain}}$ are the indices of the remaining unselected tokens.

- The fourth step is to compute the key $K_t$ and value $V_t$. This process is similar to the conventional attention computation. However, computing query $Q$ is different. Here, we only compute the queries for the selected tokens, as shown in Line 5.

- The fifth step is to perform the attention computation for the selected tokens.

---

**Algorithm 2** Evolutionary Search for Token Scheduling.

---

**Require:** $T$        ▷ Total timesteps (e.g., 100)
**Require:** $\Theta_\$$        ▷ Computation budget (e.g., total cost $\leq 50$)
**Require:** $\mathcal{P}$        ▷ Parent candidates for evaluation
**Require:** $K$        ▷ New population size
**Require:** $G$        ▷ Number of generations
**Ensure:** Best schedule $\Theta_{\text{best}} = [\theta_0, \theta_1, \ldots, \theta_{\text{T-1}}]$
1: Initialize population $\mathcal{S} = \{\Theta^{(0)}, \ldots, \Theta^{(K-1)}\}$, $\Theta^{(k)} = \{\theta_i^{(k)}\}$ with $\sum_{i=0}^{T-1} \theta_i^{(k)} \leq \Theta_\$$
2: **for** $g = 1$ to $G$ **do**
3:      **for all** $\Theta^{(k)} \in \mathcal{S}$ **do**
4:          Compute fitness: $\mathcal{L}_{\text{MSE}}^{(k)} \leftarrow \text{Evaluate}(\Theta^{(k)})$
5:      **end for**
6:      Select top individuals as parents: $\mathcal{P} \subset \mathcal{S}$
7:      Initialize new population: $\mathcal{S}_{\text{new}} \leftarrow \mathcal{P}$
8:      **while** $|\mathcal{S}_{\text{new}}| < K + |\mathcal{P}|$ **do**
9:          Sample parents $\Theta^{(a)}, \Theta^{(b)} \in \mathcal{P}$
10:         $\Theta_{\text{child}} \leftarrow \text{Crossover}(\Theta^{(a)}, \Theta^{(b)})$
11:         $\Theta_{\text{child}} \leftarrow \text{Mutate}(\Theta_{\text{child}})$
12:         $\Theta_{\text{child}} \leftarrow \text{RepairIfNeeded}(\Theta_{\text{child}}, \Theta_\$)$
13:         **if** $\Theta_{\text{child}} \notin \mathcal{S}_{\text{new}}$ **then**
14:            Add $\Theta_{\text{child}}$ to $\mathcal{S}_{\text{new}}$
15:         **end if**
16:      **end while**
17:      $\mathcal{S} \leftarrow \mathcal{S}_{\text{new}}$
18: **end for**
19: **return** $\arg\min_{\Theta \in \mathcal{S}} \text{MSE}(\Theta)$

---

- The last step is to reuse the cached token results from the previous timesteps for the unselected tokens.

## C.2 TOKEN-WISE SEARCH ALGORITHM.

Algo. 2 outlines our token-wise search algorithm via an evolutionary algorithm. This algorithm is to determine the optimal token budget allocation across denoising timesteps, as discussed in Sec. 3.3. The overall logic of this algorithm iteratively refines candidates through crossover, mutation, and repair operations to achieve a target performance with minimal accuracy loss.

**Inputs.** Our token-wise search algorithm requires four input parameters. $T$ is the total number of timesteps of the vDiT model. $\Theta_\$$ is the computational budget that we can afford for a specific performance target. $K$ is the population size of the potential candidates in the evolutionary algorithm (EA). $G$ is the total number of generations in EA.

**Output.** The output of our algorithm is the best schedule of the token selection after $G$ generations.

**Procedure.** The overall process of our evolutionary algorithm is described as follows:

- The first step is to initialize the first generation, $S$, as shown in Line 1. Each candidate $\Theta^{(k)}$ represents a token allocation schedule, which is constrained such that the total token budget across all timesteps, $\sum_{i=0}^{T-1} \theta_i^{(k)}$, does not exceed the overall token budget $\Theta_\$$.

- Then, we iterate through $G$ number of generations. For each generation, we compute the mean squared error (MSE) loss $\mathcal{L}_{\text{MSE}}^{(k)}$ between the output of the baseline model and each candidate schedule. Once we obtain the MSE loss of each candidate, we select the top candidates among $\mathcal{P}$ and obtain the subset $\mathcal{S}$.

- Then, we create the next generation of population based on $\mathcal{S}$. Here, we first initialize an empty set, $\mathcal{S}_{\text{new}}$. Next, we spawn $K$ number of new candidates based on the procedure as we describe in Sec. 3.3, following *Crossover*, *Mutate*, and *Repair* steps.

- Once we create the new set of candidates $\mathcal{S}_{\text{new}}$, we then start over the next generation utill we iterate over $G$ number of generations.

- After completing all generations, we select the candidate with the lowest MSE loss as the final output, denoted as $\Theta_{best}$.

## C.3 EXPERIMENTAL SETUP

**Hardware Platforms.** We conduct both the performance and accuracy measurements on two hardware platforms:

- NVIDIA A6000 with 38.71 TFLOPS (FP32) and 48 GB memory;
- NVIDIA A100 with 19.49 TFLOPS (FP32) and 80 GB memory.

**Video Generation Frameworks.** We measure the performance of various acceleration techniques on three widely-used video generation frameworks:HunyuanVideo-T2V (Kong et al., 2024), Wan v2.1 1.3 B Wang et al. (2025) and OpenSora v1.2 (Zheng et al., 2024). We test HunyuanVideo-T2V (Kong et al., 2024) on 5-second videos with 544x960 resolution and both Wan v2.1 1.3 B Wang et al. (2025) and OpenSora v1.2 (Zheng et al., 2024) on 2-second videos and 4-second videos with 480p resolution.

**Baselines.** We compare against five different training-free acceleration techniques:

- $\underline{\Delta\text{-D\textsc{i}T}}$ (Chen et al., 2024). The parameter $b$ represents the timestep at which the reusing strategy of the block residuals switches. The parameter $N$ represents the timestep intervals that skip full computation. In Opensora, we set $b$ and $N$ to be 15 and 3, respectively. The partial computation starts at timestep 3 and ends at timestep 28. We also preserve the residuals of the first 10 blocks or the last 10 blocks. In Wan model, we set $b$ and $N$ to be 25 and 3, respectively. The partial computation starts at timestep 5 and ends at timestep 45. We also preserve the residuals of the first 10 blocks or the last 10 blocks.

- $\underline{\text{PAB}}$ (Zhao et al., 2024). The broadcast range $n$ represents the timestep intervals that skip full computation. In the OpenSora model, the broadcast ranges of spatial attention, temporal attention, and cross-attention are set to be (2, 4, 6) or (5, 7, 9). Similarly, in the Wan model, the broadcast ranges of self-attention and cross-attention are set to be (2, 6) or (5, 9). For both models, we keep the first 15% and last 15% of the timesteps untouched.

- $\underline{\text{T\textsc{o}C\textsc{a}}}$ (Zou et al., 2024b): This variant is similar to $\underline{\text{PAB}}$. However, $\underline{\text{T\textsc{o}C\textsc{a}}}$ also performs a certain level of token-wise skipping similar to our work. We faithfully reimplement their work according to their released code (Zou et al., 2024a), which only applies the token selection on cross-attention and MLP layers. In OpenSora, their broadcast ranges of spatial attention, temporal attention, cross-attention, and MLP are 3, 3, 6, and 3, respectively. The reuse ratios of MLP in their two variants are set to 80% and 85%, respectively. In Wan model, their broadcast ranges of spatial attention, temporal attention, cross-attention, and MLP are 2, 2, 2, and 2, respectively. The reuse ratios of MLP in their two variants are also set to 80% and 85%, respectively.

- $\underline{\text{SVG}}$ (Xi et al., 2025)/$\underline{\text{SVG2}}$ (Yang et al., 2025): This variant leverages the sparsity in self-attention computation and proposes to skip some unimportant tokens during the self-attention.

**Evaluation Metrics.** Next, we describe how we obtain various performance and quality metrics.

- **Video Quality.** We used the VBench score (Huang et al., 2024) as the primary video quality metric. For each of the 946 benchmark prompts, 5 videos were generated using different random seeds. The generated videos are then evaluated across 16 aspects. We then report the average value of the aspects. The VBench score is obtained from the `VBENCH` benchmark suite (Huang et al., 2024). Specific APIs within this suite are used to evaluate aspects like `Motion Fidelity`, `Temporal Consistency`, `Aesthetic Quality`, etc.

- **Image Consistency (PSNR, SSIM, LPIPS).** To assess frame-to-frame image consistency, we compared generated videos by different acceleration methods against the original video results on image quality, using PSNR, SSIM, and LPIPS. PSNR is calculated using standard image processing libraries,

e.g., `skimage.metrics.peak_signal_noise_ratio`. Similarly, SSIM is calculated using `skimage.metrics.structural_similarity`. LPIPS is measured using a the `lpips` library in Python.

- **Performance.** For performance, we report end-to-end generation latency, GPU memory consumption, and computational complexity (FLOPs). For end-to-end latency, we measure the total time elapsed during video generation. Here, we use Python's `torch.cuda` module. We capture the start and end timestamps using `torch.cuda.event()` and use the difference between these two as the end-to-end latency. For GPU memory consumption, we use the built-in measurement to monitor the peak GPU memory usage during inference. For FLOPs, we design an analytic model to calculate the floating-point operations. Please see Sec. C .4.

## C .4 FLOPs Calculation Formulations

**Input Representation.** We first define the symbols we used in our FLOPs computation:

- $B$: Batch size.
- $N$: Number of tokens (sequence length).
- $H$: Embedding dimension (hidden dimension).
- $N_{\text{head}}$: Number of attention heads.
- $d = H/N_{\text{head}}$: Dimension per head.

**Attention FLOPs Calculation.** We abstract the computation of an attention module into two parts: linear projections for Query ($Q$), Key ($K$), and Value ($V$), and the subsequent attention score computations.

**Linear Projection.** The input $X \in \mathbb{R}^{B \times N \times H}$ is projected into $Q$, $K$, and $V$ tensors, each of shape $\mathbb{R}^{B \times N \times H}$. A single linear transformation of shape $[H \times H]$ has a FLOPs count of $2 \times B \times N \times H \times H$. Therefore, the total FLOPs for $Q$, $K$, and $V$ projections can be expressed as,

$$f_{qkv} = 3 \times (2 \times B \times N \times H \times H) = 6BNH^2. \tag{8}$$

**Attention Score Computation.** This step includes calculating the attention scores, applying softmax, computing the attention output, and output linear projection. We next describe these four parts.

1. **Compute Attention Scores ($QK^T$).** Initially, the input shapes of $Q$ and $K$ are both $[B, N_{\text{head}}, N, d]$. Their product, $QK^T$, results in a tensor of shape $[B, N_{\text{head}}, N, N]$. The FLOPs for computing $QK^T$ per head are $2 \times N \times d \times N$. If we sum across all heads and batches, the total FLOPs becomes: $FLOPs_{QK^T} = 2 \times B \times N_{\text{head}} \times N \times d \times N = 2BN_{\text{head}}N^2d$.

2. **Softmax Operation.** The softmax operation is applied to the $QK^T$ scores. Its computational cost is relatively small compared to matrix multiplications and can be approximated as: $FLOPs_{\text{softmax}} \approx B \times N_{\text{head}} \times N \times N$. For overall computational complexity, its contribution is often considered negligible.

3. **Attention Output ($A \cdot V$).** There are two inputs in this step: the attention map, $A$, which has a shape of $[B, N_{\text{head}}, N, N]$; and the value, $V$, with a shape of $[B, N_{\text{head}}, N, d]$. The attention output has a shape of $[B, N_{\text{head}}, N, d]$. The FLOPs to compute $A \cdot V$ is expressed as, $FLOPs_{\text{AV}} = 2 \times B \times N_{\text{head}} \times N^2 \times d$.

4. **Output Linear Projection (Head Merging).** Finally, the outputs from all heads are concatenated and passed through a linear layer of shape $[H \times H]$ to produce the final attention output. $FLOPs_{\text{proj}} = 2 \times B \times N \times H^2$.

**Total Self-Attention FLOPs.** By substituting $d = H/N_{\text{head}}$ and combining the above calculations, the total FLOPs for a self-attention layer simplify to,

$$FLOPs_{\text{attn-total}} = 8BNH^2 + 4BN^2H. \tag{9}$$

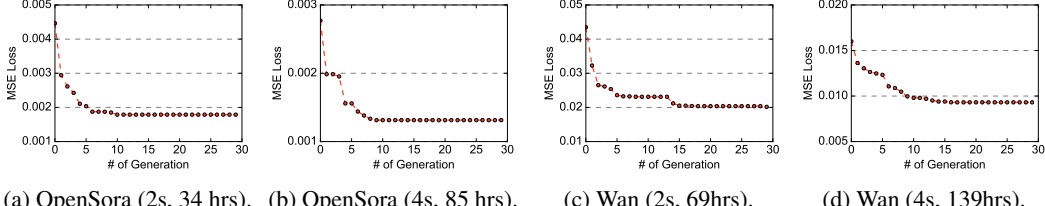

(a) OpenSora (2s, 34 hrs).    (b) OpenSora (4s, 85 hrs).    (c) Wan (2s, 69hrs).    (d) Wan (4s, 139hrs).

Fig. 11: The MSE loss trend in the EA search process. Here, we only show one case: the performance target is 50% of the token budget reduction. The trend is similar for other cases. The first number in parentheses is the video length, and the second number is the total GPU search hours.

**Cross-Attention FLOPs Calculation.** Cross-attention is similar to self-attention, but its Q and K often come from different input tokens. Here, we define $N_q$ be the number of query tokens and $N_{kv}$ be the number of key/value tokens. The FLOPs calculation can expressed as, $FLOPs_{\text{cross-attn}} = 4BN_qH^2 + 4BN_{kv}H^2 + 4BN_qN_{kv}H$.

**MLP FLOPs Calculation.** The MLP in a Transformer typically consists of two linear layers separated by an activation function (e.g., GELU). Here, we ignore the computation of the activation function and only consider the FLOPs in two linear layers. The total FLOPs for an MLP block are, $FLOPs_{\text{mlp}} = 8BNH^2 + 8BNH^2 = 16BNH^2$.

## C .5    DETAILED EXPERIMENT RESULTS

**Video Quality.** Tbl. 1 presents the overall comparison of generation quality across different techniques. On Hunyuan, ASTRAEA achieves the highest speedup (roughly 2.4×) while maintaining a better VBench score compared to other baselines. On VBench metric, all variants, ASTRAEA $_{40\%}$, ASTRAEA $_{50\%}$ and ASTRAEA $_{70\%}$, can retain the accuracy loss within 0.5%. On other metrics, ASTRAEA also achieves the best quality. For instance, ASTRAEA $_{50\%}$ achieves 1.9× speedup and has better quality metrics compared to all other methods. In contrast, the strongest baseline, SVG$_7$0%, achieves only 1.6× speedup and has much lower generative quality (e.g., PSNR, SSIM) compared to ASTRAEA $_{50\%}$. On Wan, both 2-second and 4-second video sequences, ASTRAEA $_{50\%}$ achieves a better quality in terms of all quality metrics compared to other baselines. Specifically, on the PSNR metric, ASTRAEA is almost 10 dB higher than other baselines. This shows that ASTRAEA can preserve a better quality. Similarly, on OpenSora, we can achieve almost the best accuracy while achieving the highest speedup. Although $\Delta$-DiT achieves the best VBench score on OpenSora (2s), $\Delta$-DiT can only achieve 1.01× speedup. In contrast, ASTRAEA $_{70\%}$ is almost the best on all quality metrics with much higher speedup. ASTRAEA $_{50\%}$ can achieve the second or third best on quality metrics while achieving higher speedup with a large margin. More qualitative comparisons of ASTRAEA against other methods are shown in Fig. 12, Fig. 13, and Fig. 14. Qualitatively, ASTRAEA achieves better consistency with the original models compared to other methods.

**Image consistency.** In addition to evaluating VBench scores, we also perform frame-to-frame comparisons against the outputs from the original models to assess image consistency. Our results show that ASTRAEA consistently preserves higher image consistency across all evaluated models. In particular, ASTRAEA outperforms all baseline methods by a significant margin on Wan (2s), Wan (4s), and OpenSora (2s) across all image quality metrics. For instance, on both Wan (2s) and Wan (4s), the ASTRAEA $_{70\%}$ outperforms the strongest baseline by 10 dB.

**Performance.** Tbl. 1 also shows the performance comparison across various models. ASTRAEA consistently outperforms all baselines in both inference speed and GPU memory. For Wan (2s), ASTRAEA $_{50\%}$ can deliver the highest speedup 1.9× on both A100 and A6000 with lowest memory usage. Meanwhile, it still delivers the second-best quality against the other methods. On other models, our method also achieves significantly higher speedup. Specifically, when we set the token budget to be 40%, we can achieve 2.4× speedup while the model accuracy is still competitive.

**EA Search Time.** In addition, Fig. 11 shows the EA search process of one case, 50% of the token budget reduction. Across all four evaluated models, the results converge after approximately 10 generations, indicating that a certain reduction of search time is possible. All EA searches are conducted on 8 A100 GPUs with an average search time of 82 GPU hours.

While EA might appear computationally expensive at first glance, we emphasize that EA achieves near-optimal schedules without any human-in-the-loop tuning. In contrast, prior training-free techniques, despite not requiring explicit training, rely on extensive manual effort to investigate correlations or redundancies inside models. This implicit "human training" cost is rarely accounted for, whereas EA automates this process and avoids suboptimal hand-crafted designs.

Moreover, EA can be accelerated through two complementary techniques. First, EA is inherently parallelizable: different individuals can be evaluated independently on different GPUs. Using 8 GPUs yields a 7.6× speedup, enabling the entire EA procedure to finish within one day. Second, many EA candidates share identical prefix schedules (e.g., the same decisions in the first 20 timesteps). We can evaluate those candidates together by saving the intermediate results to avoid redundant computations. Overall, we find that dynamic programming can further provide 1.5x speedup. Combining both techniques, the EA process can achieve up to 11.4× overall acceleration.

*Prompt*： *a zebra running to join a herd of its kind*

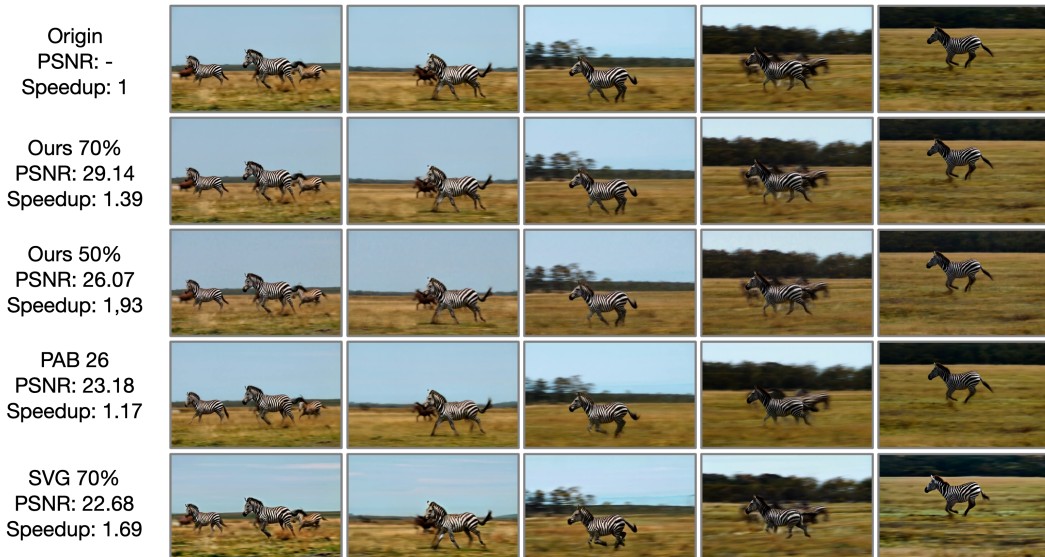

*Prompt*： *The bund Shanghai, tilt down*

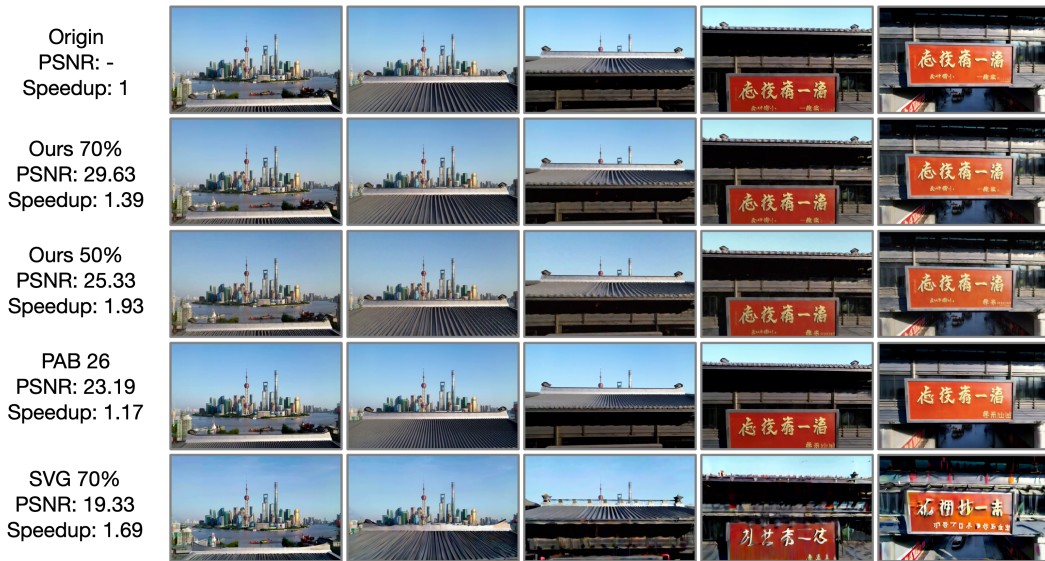

Fig. 12: The qualitative comparison of ASTRAEA against other methods on HunyuanVideo.

## C.6  FULL PERFORMANCE METRICS

In the remaining section, we show the detailed experimental results.

*Prompt*：*a raccoon dressed in suit playing the trumpet, stage background*

Origin
PSNR: -
Speedup: 1

Ours 70%
PSNR: 35.00
Speedup: 1.36

Ours 50%
PSNR: 25.76
Speedup: 1.86

PAB 26
PSNR: 18.28
Speedup: 1.37

ToCa 80%
PSNR: 24.08
Speedup: 1.60

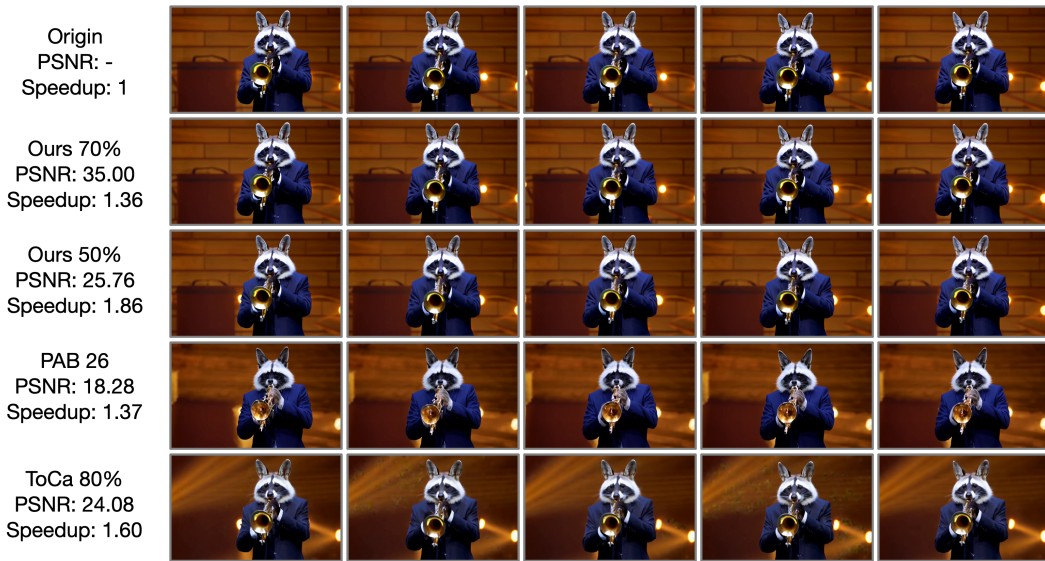

*Prompt*：*A robot DJ is playing the turnable, in heavy raining futuristic tokyo rooftop cyberpunk night, sci-fi, fantasy*

Origin
PSNR: -
Speedup: 1

Ours 70%
PSNR: 27.58
Speedup: 1.36

Ours 50%
PSNR: 23.51
Speedup: 1.86

PAB 26
PSNR: 16.37
Speedup: 1.37

ToCa 80%
PSNR: 18.87
Speedup: 1.60

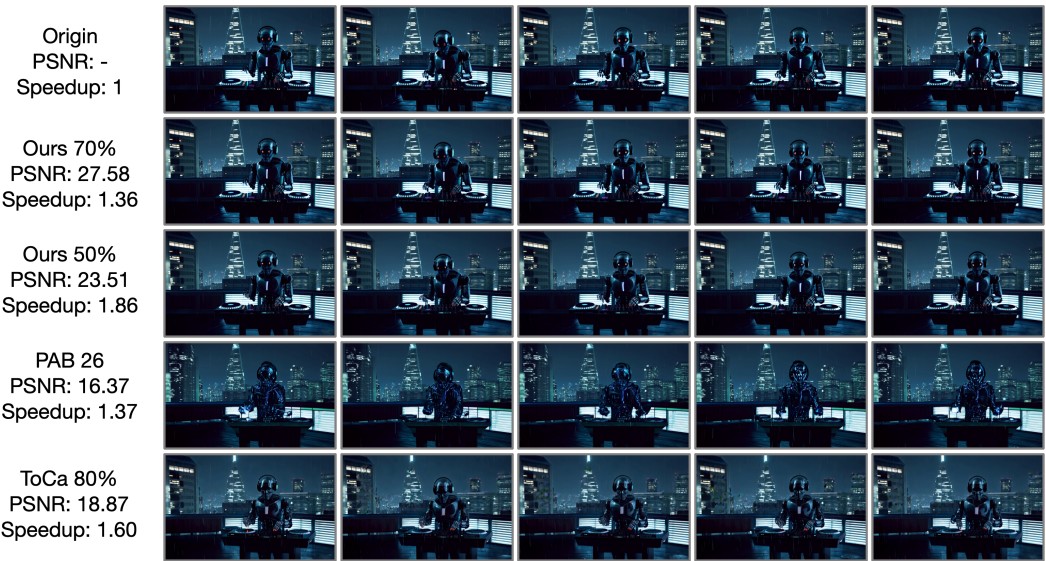

Fig. 13: The qualitative comparison of ASTRAEA against other methods on Wan (4s).

*Prompt*： *a happy fuzzy panda playing guitar nearby a campfire, snow mountain in the background*

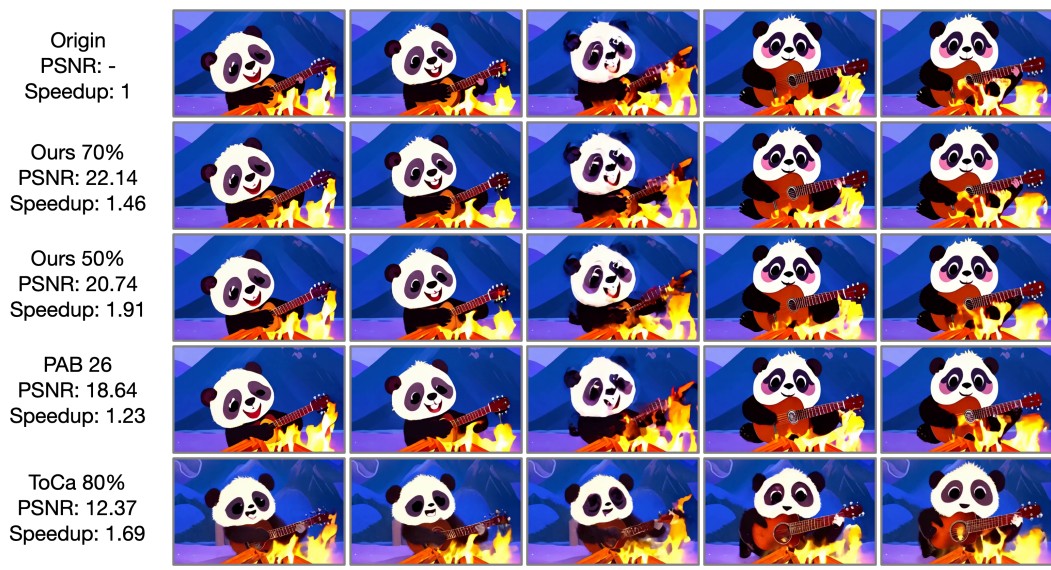

Origin
PSNR: -
Speedup: 1

Ours 70%
PSNR: 22.14
Speedup: 1.46

Ours 50%
PSNR: 20.74
Speedup: 1.91

PAB 26
PSNR: 18.64
Speedup: 1.23

ToCa 80%
PSNR: 12.37
Speedup: 1.69

*Prompt*： *A couple in formal evening wear going home get caught in a heavy downpour with umbrellas by Hokusai, in the style of Ukiyo*

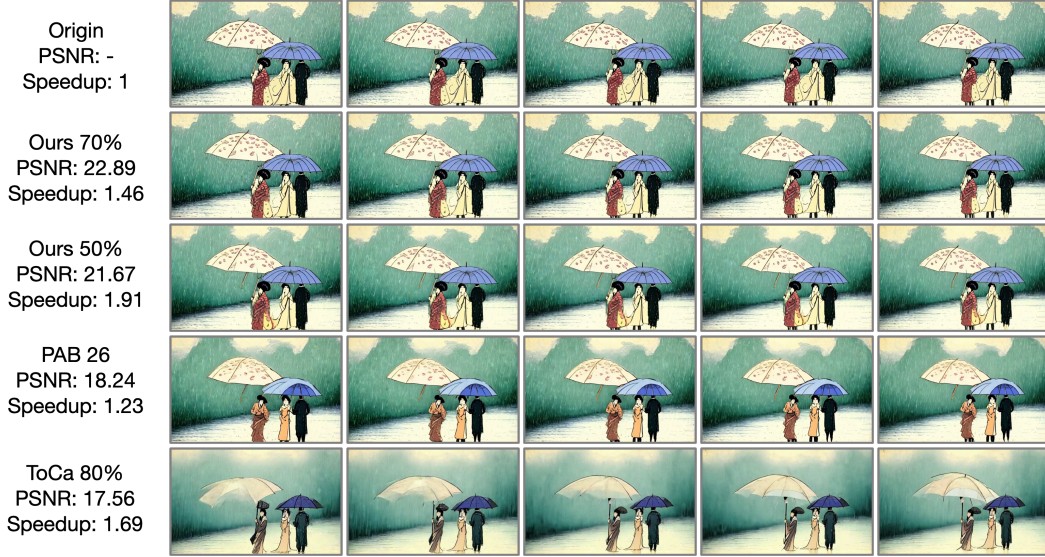

Origin
PSNR: -
Speedup: 1

Ours 70%
PSNR: 22.89
Speedup: 1.46

Ours 50%
PSNR: 21.67
Speedup: 1.91

PAB 26
PSNR: 18.24
Speedup: 1.23

ToCa 80%
PSNR: 17.56
Speedup: 1.69

Fig. 14: The qualitative comparison of ASTRAEA against other methods on OpenSora (4s).

Table 3: Individual VBench scores for HunyuanVideo model.

| Metric | Original | ASTRAEA 70% | ASTRAEA 50% | ASTRAEA 40% | SVG 70% | SVG2 70% | PAB$_{26}$ | PAB$_{59}$ | Δ-DiT |
|---|---|---|---|---|---|---|---|---|---|
| Subject Consistency | 0.9164 | 0.9236 | 0.9270 | 0.9216 | 0.9109 | 0.9622 | 0.9153 | 0.9142 | 0.9015 |
| Motion Smoothness | 0.9901 | 0.9900 | 0.9902 | 0.9901 | 0.9886 | 0.9919 | 0.9905 | 0.9915 | 0.9809 |
| Dynamic Degree | 0.8056 | 0.8571 | 0.8571 | 0.8571 | 0.7778 | 0.5000 | 0.7917 | 0.8115 | 0.7956 |
| Aesthetic Quality | 0.6234 | 0.6126 | 0.5840 | 0.5765 | 0.6181 | 0.6061 | 0.6165 | 0.6225 | 0.6206 |
| Imaging Quality | 0.6250 | 0.6494 | 0.6219 | 0.6197 | 0.6152 | 0.6524 | 0.6032 | 0.5994 | 0.6290 |
| Overall Consistency | 0.2676 | 0.2555 | 0.2551 | 0.2559 | 0.2715 | 0.2440 | 0.2663 | 0.2660 | 0.2674 |
| Background Consistency | 0.9633 | 0.9570 | 0.9668 | 0.9651 | 0.9583 | 0.9744 | 0.9620 | 0.9585 | 0.9633 |
| Object Class | 0.6305 | 0.6172 | 0.6133 | 0.6367 | 0.7112 | 0.7383 | 0.6614 | 0.6385 | 0.6915 |
| Multiple Objects | 0.5297 | 0.5469 | 0.5586 | 0.5703 | 0.5168 | 0.7930 | 0.5030 | 0.3111 | 0.5427 |
| Color | 0.8790 | 0.7641 | 0.7557 | 0.7203 | 0.8690 | 0.8934 | 0.8917 | 0.8836 | 0.8575 |
| Spatial Relationship | 0.6583 | 0.7611 | 0.7701 | 0.7705 | 0.6574 | 0.6587 | 0.6337 | 0.6470 | 0.6767 |
| Scene | 0.2885 | 0.2500 | 0.2831 | 0.2610 | 0.3009 | 0.3787 | 0.2696 | 0.1322 | 0.2929 |
| Temporal Style | 0.2367 | 0.2350 | 0.2359 | 0.2337 | 0.2421 | 0.2371 | 0.2357 | 0.2333 | 0.2371 |
| Human Action | 0.8900 | 0.8500 | 0.9000 | 0.8500 | 0.9200 | 0.9500 | 0.8400 | 0.8200 | 0.8800 |
| Temporal Flickering | 0.9861 | 0.9872 | 0.9874 | 0.9874 | 0.9852 | 0.9926 | 0.9869 | 0.9883 | 0.9761 |
| Appearance Style | 0.1908 | 0.1826 | 0.1841 | 0.1827 | 0.1942 | 0.1972 | 0.1901 | 0.1896 | 0.1898 |
| Quality Score | 0.8371 | 0.8435 | 0.8377 | 0.8348 | 0.8294 | 0.8293 | 0.8316 | 0.8336 | 0.8373 |
| Semantic Score | 0.6657 | 0.6477 | 0.6594 | 0.6499 | 0.6812 | 0.7228 | 0.6557 | 0.6107 | 0.6727 |
| Total Score | 0.8028 | 0.8043 | 0.8020 | 0.7979 | 0.7997 | 0.8080 | 0.7964 | 0.7890 | 0.7943 |

Table 4: Individual VBench scores for Wan (4s) model.

| Metric | Original | ASTRAEA 70% | ASTRAEA 50% | ASTRAEA 40% | ToCa 80% | ToCa 85% | SVG 70% | SVG2 | PAB$_{26}$ | PAB$_{59}$ | Δ-DiT |
|---|---|---|---|---|---|---|---|---|---|---|---|
| Subject Consistency | 0.9576 | 0.9579 | 0.9585 | 0.9591 | 0.9478 | 0.9480 | 0.9374 | 0.9402 | 0.9556 | 0.9557 | 0.9510 |
| Motion Smoothness | 0.9826 | 0.9828 | 0.9830 | 0.9832 | 0.9816 | 0.9821 | 0.9700 | 0.9776 | 0.9831 | 0.9839 | 0.9802 |
| Dynamic Degree | 0.6389 | 0.6389 | 0.6250 | 0.6111 | 0.5694 | 0.5833 | 0.7857 | 0.7500 | 0.5694 | 0.5139 | 0.5714 |
| Aesthetic Quality | 0.6116 | 0.6123 | 0.6162 | 0.6169 | 0.5966 | 0.6004 | 0.5544 | 0.5887 | 0.5921 | 0.5837 | 0.5566 |
| Imaging Quality | 0.6410 | 0.6392 | 0.6331 | 0.6316 | 0.6348 | 0.6363 | 0.6285 | 0.6294 | 0.6387 | 0.6214 | 0.5730 |
| Overall Consistency | 0.2361 | 0.2362 | 0.2355 | 0.2343 | 0.2396 | 0.2387 | 0.2227 | 0.2299 | 0.2248 | 0.2191 | 0.2291 |
| Background Consistency | 0.9888 | 0.9888 | 0.9892 | 0.9894 | 0.9668 | 0.9683 | 0.9481 | 0.9714 | 0.9901 | 0.9901 | 0.9798 |
| Object Class | 0.7587 | 0.7658 | 0.7619 | 0.7611 | 0.7682 | 0.7682 | 0.6367 | 0.7239 | 0.7650 | 0.7350 | 0.7969 |
| Multiple Objects | 0.5663 | 0.5518 | 0.5450 | 0.5396 | 0.5450 | 0.5587 | 0.4219 | 0.5648 | 0.4482 | 0.4002 | 0.3516 |
| Color | 0.8754 | 0.8751 | 0.8715 | 0.8895 | 0.8990 | 0.9079 | 0.9042 | 0.8547 | 0.8709 | 0.8786 | 0.8438 |
| Spatial Relationship | 0.7286 | 0.7224 | 0.7001 | 0.6841 | 0.7717 | 0.7846 | 0.6489 | 0.7167 | 0.6608 | 0.6171 | 0.7447 |
| Scene | 0.2594 | 0.2485 | 0.2485 | 0.2238 | 0.2347 | 0.2166 | 0.1544 | 0.1330 | 0.2122 | 0.2064 | 0.1066 |
| Temporal Style | 0.2416 | 0.2408 | 0.2394 | 0.2384 | 0.2451 | 0.2445 | 0.2276 | 0.2319 | 0.2269 | 0.2172 | 0.2403 |
| Human Action | 0.7400 | 0.7300 | 0.7300 | 0.7300 | 0.7400 | 0.7500 | 0.7500 | 0.7400 | 0.6800 | 0.6700 | 0.6500 |
| Temporal Flickering | 0.9943 | 0.9938 | 0.9929 | 0.9924 | 0.9903 | 0.9910 | 0.9898 | 0.9915 | 0.9907 | 0.9929 | 0.9898 |
| Appearance Style | 0.1992 | 0.1988 | 0.1987 | 0.1982 | 0.1995 | 0.1991 | 0.2239 | 0.2074 | 0.1979 | 0.1986 | 0.2046 |
| Quality Score | 0.8370 | 0.8368 | 0.8353 | 0.8342 | 0.8199 | 0.8227 | 0.8170 | 0.8297 | 0.8284 | 0.8202 | 0.8067 |
| Semantic Score | 0.6660 | 0.6614 | 0.6567 | 0.6521 | 0.6710 | 0.6730 | 0.6189 | 0.6396 | 0.6240 | 0.6050 | 0.6135 |
| Total Score | 0.8028 | 0.8018 | 0.7996 | 0.7978 | 0.7901 | 0.7928 | 0.7774 | 0.7917 | 0.7876 | 0.7771 | 0.7681 |

Table 5: Individual VBench scores for Wan (2s) model.

| Metric | Original | ASTRAEA 70% | ASTRAEA 50% | ASTRAEA 40% | ToCa 80% | ToCa 85% | SVG 70% | SVG2 70% | PAB$_{26}$ | PAB$_{59}$ | Δ-DiT |
|---|---|---|---|---|---|---|---|---|---|---|---|
| Subject Consistency | 0.9719 | 0.9722 | 0.9719 | 0.9726 | 0.9575 | 0.9506 | 0.9511 | 0.9513 | 0.9705 | 0.9684 | 0.9605 |
| Motion Smoothness | 0.9833 | 0.9832 | 0.9832 | 0.9816 | 0.9811 | 0.9819 | 0.9721 | 0.9768 | 0.9835 | 0.9836 | 0.9779 |
| Dynamic Degree | 0.5972 | 0.5833 | 0.5833 | 0.5833 | 0.7143 | 0.6389 | 0.7143 | 0.8333 | 0.6250 | 0.6250 | 0.5417 |
| Aesthetic Quality | 0.6279 | 0.6272 | 0.6278 | 0.6351 | 0.6142 | 0.6208 | 0.5386 | 0.5807 | 0.6793 | 0.6630 | 0.5906 |
| Imaging Quality | 0.6801 | 0.6788 | 0.6773 | 0.6642 | 0.6723 | 0.6717 | 0.6390 | 0.6421 | 0.6793 | 0.6630 | 0.6463 |
| Overall Consistency | 0.2383 | 0.2378 | 0.2386 | 0.2370 | 0.2175 | 0.2409 | 0.2153 | 0.2245 | 0.2266 | 0.2198 | 0.2335 |
| Background Consistency | 0.9884 | 0.9889 | 0.9887 | 0.9897 | 0.9656 | 0.9662 | 0.9629 | 0.9703 | 0.9885 | 0.9884 | 0.9789 |
| Object Class | 0.7595 | 0.7634 | 0.7832 | 0.7627 | 0.8906 | 0.8022 | 0.7148 | 0.7453 | 0.7381 | 0.6843 | 0.7191 |
| Multiple Objects | 0.6654 | 0.6700 | 0.6441 | 0.6159 | 0.4492 | 0.6601 | 0.3906 | 0.4794 | 0.4192 | 0.3377 | 0.4710 |
| Color | 0.9188 | 0.8857 | 0.8714 | 0.9350 | 0.8978 | 0.8705 | 0.8095 | 0.8440 | 0.8662 | 0.8431 | 0.8227 |
| Spatial Relationship | 0.7988 | 0.8023 | 0.7888 | 0.7441 | 0.7573 | 0.8283 | 0.5690 | 0.6221 | 0.7446 | 0.6400 | 0.7368 |
| Scene | 0.3089 | 0.3045 | 0.3067 | 0.2907 | 0.3971 | 0.3743 | 0.2132 | 0.1453 | 0.2871 | 0.2536 | 0.2304 |
| Temporal Style | 0.2345 | 0.2337 | 0.2328 | 0.2309 | 0.2285 | 0.2322 | 0.2328 | 0.2318 | 0.2175 | 0.2061 | 0.2202 |
| Human Action | 0.7800 | 0.7900 | 0.7600 | 0.7900 | 0.8000 | 0.7500 | 0.7000 | 0.7100 | 0.6800 | 0.5900 | 0.7200 |
| Temporal Flickering | 0.9900 | 0.9893 | 0.9887 | 0.9853 | 0.9848 | 0.9862 | 0.9900 | 0.9939 | 0.9907 | 0.9891 | 0.9876 |
| Appearance Style | 0.1994 | 0.1985 | 0.1981 | 0.1986 | 0.1962 | 0.2005 | 0.2223 | 0.2070 | 0.1951 | 0.1956 | 0.2049 |
| Quality Score | 0.8434 | 0.8418 | 0.8414 | 0.8385 | 0.8385 | 0.8335 | 0.8174 | 0.8392 | 0.8422 | 0.8360 | 0.8203 |
| Semantic Score | 0.6994 | 0.6968 | 0.6898 | 0.6868 | 0.6991 | 0.7105 | 0.6058 | 0.6173 | 0.6338 | 0.5867 | 0.6371 |
| Total Score | 0.8146 | 0.8128 | 0.8111 | 0.8082 | 0.8106 | 0.8089 | 0.7751 | 0.7948 | 0.8005 | 0.7861 | 0.7837 |

Table 6: Individual VBench scores for OpenSora (4s) model.

| Metric | Original | ASTRAEA 70% | ASTRAEA 50% | ASTRAEA 40% | ToCa 80% | ToCa 85% | PAB$_{246}$ | PAB$_{579}$ | $\Delta$-DiT |
|---|---|---|---|---|---|---|---|---|---|
| Subject Consistency | 0.9478 | 0.9471 | 0.9459 | 0.9340 | 0.9475 | 0.9462 | 0.9482 | 0.9360 | 0.9504 |
| Motion Smoothness | 0.9851 | 0.9872 | 0.9865 | 0.9751 | 0.9844 | 0.9842 | 0.9885 | 0.9889 | 0.9817 |
| Dynamic Degree | 0.5417 | 0.5000 | 0.4722 | 0.4861 | 0.3750 | 0.3750 | 0.4583 | 0.4028 | 0.4028 |
| Aesthetic Quality | 0.5560 | 0.5533 | 0.5483 | 0.5315 | 0.5637 | 0.5633 | 0.5509 | 0.5322 | 0.5601 |
| Imaging Quality | 0.5932 | 0.5831 | 0.5750 | 0.5498 | 0.5535 | 0.5537 | 0.5783 | 0.5509 | 0.5974 |
| Overall Consistency | 0.2742 | 0.2734 | 0.2718 | 0.2656 | 0.2716 | 0.2720 | 0.2734 | 0.2611 | 0.2635 |
| Background Consistency | 0.9751 | 0.9745 | 0.9718 | 0.9699 | 0.9695 | 0.9700 | 0.9713 | 0.9637 | 0.9767 |
| Object Class | 0.8062 | 0.8014 | 0.7903 | 0.8030 | 0.8687 | 0.8552 | 0.7967 | 0.7856 | 0.8972 |
| Multiple Objects | 0.4977 | 0.4947 | 0.4703 | 0.4566 | 0.5213 | 0.5358 | 0.5137 | 0.4764 | 0.5793 |
| Color | 0.7925 | 0.8047 | 0.8221 | 0.7814 | 0.8806 | 0.8659 | 0.8203 | 0.7871 | 0.8179 |
| Spatial Relationship | 0.6326 | 0.6149 | 0.6036 | 0.5724 | 0.6168 | 0.6308 | 0.6035 | 0.5979 | 0.5705 |
| Scene | 0.4135 | 0.4302 | 0.4215 | 0.4208 | 0.3874 | 0.3917 | 0.4484 | 0.3953 | 0.4564 |
| Temporal Style | 0.2412 | 0.2406 | 0.2390 | 0.2351 | 0.2481 | 0.2481 | 0.2393 | 0.2319 | 0.2384 |
| Human Action | 0.8800 | 0.8800 | 0.8700 | 0.8600 | 0.8600 | 0.8500 | 0.8800 | 0.8400 | 0.8900 |
| Temporal Flickering | 0.9952 | 0.9951 | 0.9946 | 0.9922 | 0.9946 | 0.9947 | 0.9951 | 0.9946 | 0.9937 |
| Appearance Style | 0.2380 | 0.2379 | 0.2374 | 0.2357 | 0.2381 | 0.2394 | 0.2377 | 0.2357 | 0.2361 |
| Quality Score | 0.8107 | 0.8064 | 0.8009 | 0.7859 | 0.7911 | 0.7909 | 0.8023 | 0.7871 | 0.7997 |
| Semantic Score | 0.7068 | 0.7071 | 0.7003 | 0.6873 | 0.7200 | 0.7202 | 0.7111 | 0.6830 | 0.7239 |
| Total Score | 0.7899 | 0.7865 | 0.7807 | 0.7662 | 0.7769 | 0.7768 | 0.7840 | 0.7663 | 0.7846 |

Table 7: Individual VBench scores for OpenSora (2s) model.

| Metric | Original | ASTRAEA 70% | ASTRAEA 50% | ASTRAEA 40% | ToCa 80% | ToCa 85% | PAB$_{246}$ | PAB$_{579}$ | $\Delta$-DiT |
|---|---|---|---|---|---|---|---|---|---|
| Subject Consistency | 0.9664 | 0.9675 | 0.9682 | 0.9615 | 0.9576 | 0.9570 | 0.9662 | 0.9591 | 0.9615 |
| Motion Smoothness | 0.9845 | 0.9864 | 0.9878 | 0.9826 | 0.9854 | 0.9853 | 0.9875 | 0.9885 | 0.9802 |
| Dynamic Degree | 0.3333 | 0.2917 | 0.3056 | 0.3611 | 0.3194 | 0.2917 | 0.2500 | 0.4286 | 0.4444 |
| Aesthetic Quality | 0.5681 | 0.5684 | 0.5676 | 0.5582 | 0.5592 | 0.5584 | 0.5683 | 0.5398 | 0.5463 |
| Imaging Quality | 0.5992 | 0.5930 | 0.5827 | 0.5771 | 0.5545 | 0.5554 | 0.5798 | 0.5376 | 0.5550 |
| Overall Consistency | 0.2722 | 0.2721 | 0.2701 | 0.2698 | 0.2723 | 0.2724 | 0.2709 | 0.2624 | 0.2469 |
| Background Consistency | 0.9790 | 0.9781 | 0.9734 | 0.9753 | 0.9717 | 0.9691 | 0.9766 | 0.9630 | 0.9738 |
| Object Class | 0.8347 | 0.8402 | 0.8402 | 0.8354 | 0.8473 | 0.8544 | 0.8576 | 0.8331 | 0.9531 |
| Multiple Objects | 0.4177 | 0.4238 | 0.4040 | 0.4070 | 0.4451 | 0.4261 | 0.4200 | 0.3636 | 0.6602 |
| Color | 0.7947 | 0.8013 | 0.7762 | 0.7693 | 0.7373 | 0.7539 | 0.7383 | 0.7995 | 0.7538 |
| Spatial Relationship | 0.5854 | 0.5779 | 0.5702 | 0.5673 | 0.5225 | 0.5381 | 0.5793 | 0.5231 | 0.4717 |
| Scene | 0.4295 | 0.4215 | 0.3910 | 0.3903 | 0.4331 | 0.4331 | 0.4368 | 0.3474 | 0.5441 |
| Temporal Style | 0.2470 | 0.2466 | 0.2449 | 0.2442 | 0.2455 | 0.2455 | 0.2435 | 0.2351 | 0.2389 |
| Human Action | 0.8600 | 0.8700 | 0.8700 | 0.8600 | 0.8400 | 0.8500 | 0.8400 | 0.8300 | 0.9000 |
| Temporal Flickering | 0.9947 | 0.9946 | 0.9947 | 0.9932 | 0.9942 | 0.9943 | 0.9946 | 0.9940 | 0.9931 |
| Appearance Style | 0.2407 | 0.2402 | 0.2397 | 0.2385 | 0.2420 | 0.2423 | 0.2403 | 0.2384 | 0.2406 |
| Quality Score | 0.8022 | 0.8012 | 0.7962 | 0.7908 | 0.7922 | 0.7883 | 0.7960 | 0.7780 | 0.7934 |
| Semantic Score | 0.6982 | 0.6991 | 0.6878 | 0.6845 | 0.6876 | 0.6912 | 0.6912 | 0.6637 | 0.7308 |
| Total Score | 0.7814 | 0.7808 | 0.7745 | 0.7695 | 0.7713 | 0.7689 | 0.7750 | 0.7552 | 0.7809 |

Table 8: FLOPs Breakdown for HunyuanVideo model across different methods (in $10^{15}$ FLOPs).

| Method | Self | Cross | MLP |
|---|---|---|---|
| **Original** | 168.34 | - | 45.93 |
| **Delta-DiT** | 107.73 | - | 37.36 |
| **PAB 26** | 131.31 | - | 45.09 |
| **PAB 59** | 101.01 | - | 44.55 |
| **SVG 70%** | 75.58 | - | 45.93 |
| **ASTRAEA 0.4** | 67.34 | - | 45.93 |
| **ASTRAEA 0.5** | 84.17 | - | 45.93 |
| **ASTRAEA 0.7** | 117.84 | - | 45.93 |

Table 9: FLOPs Breakdown for Wan (2s) model across different methods (in $10^{15}$ FLOPs).

| Method | Self | Cross | MLP |
|---|---|---|---|
| **Original** | 3.6830 | 0.1491 | 1.5900 |
| **Delta-DiT** | 2.9464 | 0.1192 | 1.2720 |
| **ToCa 0.8** | 1.9520 | 0.0790 | 0.9921 |
| **ToCa 0.85** | 1.9520 | 0.0790 | 0.9548 |
| **PAB 246** | 2.3940 | 0.0621 | 1.5900 |
| **PAB 579** | 1.6205 | 0.0563 | 1.5900 |
| **ASTRAEA 0.4** | 1.4732 | 0.0596 | 0.6360 |
| **ASTRAEA 0.5** | 1.8415 | 0.0745 | 0.7950 |
| **ASTRAEA 0.7** | 2.5781 | 0.1043 | 1.1130 |

Table 10: FLOPs Breakdown for Wan (4s) model across different methods (in $10^{15}$ FLOPs).

| Method | Self | Cross | MLP |
|--------|------|-------|-----|
| **Original** | 13.0573 | 0.2816 | 3.0033 |
| **Delta-DiT** | 10.4458 | 0.2252 | 2.4026 |
| **ToCa 0.8** | 6.9204 | 0.1492 | 1.8741 |
| **ToCa 0.85** | 6.9204 | 0.1492 | 1.8035 |
| **PAB 246** | 8.4872 | 0.1173 | 3.0033 |
| **PAB 579** | 5.7452 | 0.1064 | 3.0033 |
| **ASTRAEA 0.4** | 5.2229 | 0.1126 | 1.2013 |
| **ASTRAEA 0.5** | 6.5286 | 0.1408 | 1.5016 |
| **ASTRAEA 0.7** | 9.1401 | 0.1971 | 2.1023 |

Table 11: FLOPs Breakdown for OpenSora (2s) model across different methods (in $10^{15}$ FLOPs).

| Method | Spatial | Temporal | Cross | MLP |
|--------|---------|----------|-------|-----|
| **Original** | 0.7190 | 0.4282 | 0.4380 | 0.8508 |
| **Delta-DiT** | 0.5512 | 0.3283 | 0.3358 | 0.6523 |
| **ToCa 0.8** | 0.2876 | 0.1713 | 0.3504 | 0.4424 |
| **ToCa 0.85** | 0.2450 | 0.1287 | 0.1802 | 0.4169 |
| **PAB 246** | 0.4673 | 0.2034 | 0.1825 | 0.8508 |
| **PAB 579** | 0.3163 | 0.1713 | 0.1654 | 0.8508 |
| **ASTRAEA 0.4** | 0.2876 | 0.1713 | 0.1752 | 0.3403 |
| **ASTRAEA 0.5** | 0.3595 | 0.2141 | 0.2190 | 0.4254 |
| **ASTRAEA 0.7** | 0.5033 | 0.2997 | 0.3066 | 0.5956 |

Table 12: FLOPs Breakdown for OpenSora (4s) model across different methods (in $10^{15}$ FLOPs).

| Method | Spatial | Temporal | Cross | MLP |
|--------|---------|----------|-------|-----|
| **Original** | 1.4379 | 0.8619 | 0.8759 | 1.7016 |
| **Delta-DiT** | 1.1024 | 0.6608 | 0.6715 | 1.3046 |
| **ToCa 0.8** | 0.5752 | 0.3447 | 0.7007 | 0.8848 |
| **ToCa 0.85** | 0.2450 | 0.1287 | 0.1802 | 0.8338 |
| **PAB 246** | 0.9347 | 0.4094 | 0.3650 | 1.7016 |
| **PAB 579** | 0.6327 | 0.3447 | 0.3309 | 1.7016 |
| **ASTRAEA 0.4** | 0.5752 | 0.3447 | 0.3504 | 0.6806 |
| **ASTRAEA 0.5** | 0.7190 | 0.4309 | 0.4380 | 0.8508 |
| **ASTRAEA 0.7** | 1.0065 | 0.6033 | 0.6131 | 1.1911 |

