# OpenReview forum: "Astraea: A Token-wise Acceleration Framework for Video Diffusion Transformers"
_ICLR.cc/2026/Conference — ICLR 2026 Poster_

### Official Review · Reviewer_hZdk · 2025-10-24

**Soundness:** 2
**Presentation:** 2
**Contribution:** 2
**Rating:** 4
**Confidence:** 4

**Summary:**

This paper proposed ASTRAEA, a token-wise cache framework for efficient video generation. ASTRAEA proposes a lightweight token selection mechanism and a memory-efficient, GPU-friendly sparse attention strategy. ASTRAEA also proposed an evolutionary algorithm to automatically determine the distribution of the token budget across timesteps. Experiments show that ASTRAEA achieves better generation quality with lower latency under different video diffusion transformers.

**Strengths:**

1.The problem studied in the paper is quite valuable, and the acceleration of video DiT generation is crucial.

2.ASTRAEA should be able to be migrated to various DiT architectures with good generalization.

3.The method explanation is relatively clear and easy to understand.

4.The performance and acceleration effect of ASTRAEA appear to be relatively good.

**Weaknesses:**

1.The acquisition of LSE Score should not be directly provided by existing attention methods such as FlashAttention. How is the efficient and memory friendly acquisition described in the paper achieved? Can specific resource consumption be reported?

2.The sparse attention in Fig.3 appears to bring additional computational complexity, why does the paper claim less computational complexity compared to sparsification for QK? Sparse QK theoretically should be faster, and faster hardware implementation can be achieved using a kernel designed like in SVG. Can you provide a more detailed explanation.

3.The evolutionary algorithm used in the paper appears to be very time-consuming, which puts it at a certain disadvantage compared to other training-free methods. Can you report the performance comparison without using the evolutionary algorithm?

4.The paper lacks the performance analysis of individual components.

5.It seems that the statement in line 463 is incorrect and contradictory to line 464.

6.What are the hyperparameters set in Eq.4 and how should they be set? The paper lacks additional analysis on whether these hyperparameters are sensitive.

**Questions:**

Please see above weaknesses.

---

> ### Author Response · Authors · 2025-11-21
>
> Thanks for your valuable feedback!
>
> ## Q1: How to obtain the LSE score from FlashAttention?
>
> Current FlashAttention implementations (both v2 and v3) provide a dedicated interface to obtain the LSE score used internally for numerical stability. Specifically, you can obtain the LSE score via the official API by setting “return_attn_probs=True”. It will return the LSE score.
>
> Inside the FlashAttention kernel, it computes LSE in the fused kernel during the streaming softmax. So it requires ZERO extra computation and minimal memory overhead.
>
> ## Q2: Sparse attention in Fig. 3 seems to add computation. Why claim lower complexity than QK sparsification? Sparse QK should be faster.
>
> Nowhere in Fig. 3 or throughout Section 3.2, we state or indicate “our sparse attention computation has lower complexity than QK sparsification.” We only claim that “our Q-only sparse attention is more GPU-friendly than QK sparsification” and “our approach saves less computation than naive sparse attention (QK sparsification)”.
>
> There are three reasons that our attention computation is more GPU-friendly (as we already stated in the paper):
> 1. It does not require storing the entire attention scores from the previous timestep to fill the missing elements in the attention map. Storing the entire attention scores almost doubles the GPU memory compared to the baseline (see Table 1).
> 2. All GPUs adopt a SIMT (single-instruction-multiple-thread) execution fashion. Unstructured QK sparsity (like our case) will achieve almost no speedup compared to Q-only sparsity.
> 3. Lastly, QK-sparsity requires filling in missing attention scores in the attention map. It would introduce irregular memory accesses, which would further slow down the performance. ToCa in Table 1 uses QK sparsification. It is much slower than ours.
>
> ## Q3: Sparse QK theoretically should be faster. Can a faster hardware implementation be achieved using a kernel designed like SVG?
>
> Very unlikely. On modern GPUs, accelerating irregular, unstructured sparse attention computation is always an open question. Those sparse attention computations often have low GPU utilization. To improve the GPU utilization, people often transform those sparse attentions into somewhat dense attention computation using some offline shuffling strategies, like SVG. However, our method dynamically selects reused tokens at runtime. Those offline strategies won’t help improve the performance.
>
> Additionally, sparse QK requires filling in the missing scores in the attention map. The memory access overheads are non-trivial. (See Q2)
>
> ## Q4: EA is time-consuming. Can you show performance without EA?
>
> Although EA is time-consuming, we can accelerate it via multi-GPU parallelism (7.6x speedup) and dynamic programming (1.5x speedup). Overall, we can achieve 11.4x speedup on 8 GPUs.
>
> Our EA result also generalizes well across different vDiT models. Here, we demonstrate that the search results obtained from HunyuanVideo can directly apply to Wan 2.1.
>
> |VBench Metric|Astraea 50%|Astraea 50% from HunyuanVideo|
> |-|-|-|
> |quality score|0.835|0.842|
> |semantic score|0.657|0.620|
> |total score|0.800|0.798|
> |PSNR|28.10|27.75|
>
> Overall, we can achieve a similar generative quality to that obtained by performing EA dedicatedly on Wan 2.1.
>
> ## Q5: Missing performance analysis of individual components.
>
> We provide the performance breakdown of ASTRAEA-50% on the three models (Wan2.1 1.3B, HunyuanVideo, OpenSora-1.2).
>
> Wan2.1 1.3B 65frame 480p, unit: second.
> |Module|token select|self attn|cross attn|mlp|other|total|
> |-|-|-|-|-|-|-|
> |Astraea 50%|1.50|55.32|8.21|14.92|3.39|83.35|
> |org|0.00|105.72|15.12|28.03|6.16|155.03|
>
> HunyuanVideo 129frame 544x960, unit: second.
> |Mode|token select|self attn|cross attn|mlp|other|total|
> |-|-|-|-|-|-|-|
> |Astraea 50%|13.23|454.29|—|73.77|94.58|635.87|
> |org|0.00|906.09|—|146.57|174.00|1226.66|
>
> OpenSora-1.2 93frame 480p, unit: second.
> |Mode|token select|self attn|cross attn|mlp|other|total|
> |-|-|-|-|-|-|-|
> |Astraea 50%|1.86|22.82|9.93|19.88|4.12|58.61|
> |org|0.00|43.32|19.53|38.36|7.82|109.02|
>
> ## Q6: Statement in line 463 contradicts line 464.
>
> Yes, it is indeed a typo, and we fixed it.
>
> ## Q7: What are the hyperparameters in Eq. 4, and is the method sensitive to them?
>
> We performed a sensitivity study for hyperparameters in Eqn. 4:
> $W_{\alpha}$: significance term; $W_{\beta}$: penalty term for non-selected tokens.
>
> Here, we fix $W_{\alpha} = 1$ and only vary $W_{\beta}$. The result below is for Wan2.1.
>
> |$W_\beta$|0.1|0.3|0.5|0.7|0.9|1.5|
> |-|-|-|-|-|-|-|
> |PSNR|25.63|25.65|25.67|25.64|25.60|25.46|
>
> The results show: $W_{\alpha}=1,W_{\beta}=0.5$ is the optimal setting. The overall variation is extremely small (<0.2 PSNR), indicating that our method is robust to these hyperparameters.
> Also, we would like to emphasize that this penalty term is adapted directly from ToCa, and our sensitivity trend matches the trend reported in ToCa. Thus, we claim no contribution to this term, as mentioned in the paper.

---

> > ### Comment · Reviewer_hZdk · 2025-11-25
> >
> > Thanks for the reply.
> > ﻿
> >
> > However, my biggest concern still lies within the EA search time, which the authors claim costs 82GPU hours with 8 A100 GPUs. Can you demonstrate some performance comparisons without EA? Or does the main performance improvement still come from EA, which is an unavoidable trade-off?

---

> > > ### Author Response · Authors · 2025-11-26
> > >
> > > ## Q1: EA is time-consuming.
> > >
> > > First, we have to clarify an important point: the reported search time of 82 GPU hours refers to the total search time, i.e., we only require roughly **10 hours** on a cluster of 8 A100 GPUs. With the additional memoization technique, we can further reduce the search time to **7.2 hours** (1.5x speedup). Moreover, one search result can be reused across millions of inference services. Given our collaboration with one of the leading cloud service providers in the world, ``half a day’’ search time is negligible, especially considering the substantial gains in quality and performance from our approach.
> > >
> > > Second, I don’t believe all prior work is free of cost at all. Although they do not have an explicit search process, they all require laborious experiments to identify potential correlations/redundancies within each vDiT model. Identifying those hyperparameters used in their techniques is still a time-consuming process and has not been explicitly counted into their “human search” process. Our EA algorithm simply automates this “human search” process in a more elegant way and avoids suboptimal solutions in prior work.
> > >
> > > Lastly, we show that one EA result from a single model is portable across models, i.e., this EA result can be used on other vDiT models. Here, we show that a schedule searched on HunyuanVideo can transfer to Wan2.1 and OpenSora. Note that, the total timestep of OpenSora is 30. Thus, we linearly downsample the Hunyuan search result from 50 to 30 timesteps and apply it to OpenSora. This way, the search time is further amortized to **1.4 hours**. We DO NOT think 1.4 hours is “time-consuming” at all.
> > >
> > > The overall results are shown below (we only show the results from 4-second videos):
> > > |Metric|OpenSora Astraea 50%|OpenSora Astraea 50% using HunyuanVideo EA result|Wan Astraea 50%|Wan Astraea 50% using HunyuanVideo EA result|
> > > |-|-|-|-|-|
> > > |Subject Consistency|0.946|0.936|0.959|0.958|
> > > |Motion Smoothness|0.987|0.978|0.983|0.984|
> > > |Dynamic Degree|0.472|0.643|0.625|0.643|
> > > |Aesthetic Quality|0.548|0.552|0.616|0.647|
> > > |Imaging Quality|0.575|0.559|0.633|0.642|
> > > |Overall Consistency|0.272|0.264|0.236|0.223|
> > > |Background Consistency|0.972|0.957|0.989|0.986|
> > > |Object Class|0.790|0.785|0.762|0.762|
> > > |Multiple Objects|0.470|0.504|0.545|0.465|
> > > |Color|0.822|0.852|0.872|0.745|
> > > |Spatial Relationship|0.604|0.409|0.700|0.718|
> > > |Scene|0.422|0.452|0.249|0.235|
> > > |Temporal Style|0.239|0.235|0.239|0.250|
> > > |Human Action|0.870|0.800|0.730|0.600|
> > > |Temporal Flickering|0.995|0.995|0.993|0.991|
> > > |Appearance Style|0.237|0.233|0.199|0.203|
> > > |Quality Score|0.801|0.803|0.835|0.842|
> > > |Semantic Score|0.700|0.676|0.657|0.620|
> > > |Total Score|0.781|0.778|0.800|0.798|
> > > |PSNR|28.51|24.69|28.12|27.75|
> > >
> > > Overall, using a schedule from other models would decrease the generative quality. However, it is still significantly better than prior work.
> > >
> > > ## Q2: The performance without EA.
> > >
> > >
> > > For a fair comparison, we include a heuristic baseline **without EA**, which uses a fixed token computation ratio and performs a full computation every 3 timesteps. The resulting performance remains noticeably worse than the EA-optimized schedules under the same compute budget. This shows that EA brings substantial quality improvements that cannot be matched by simple heuristics.
> > > |Metric|Astraea 50%|FixedRatio 50%|
> > > |-|-|-|
> > > |Subject Consistency|0.959|0.946|
> > > |Motion Smoothness|0.983|0.988|
> > > |Dynamic Degree|0.625|0.500|
> > > |Aesthetic Quality|0.616|0.563|
> > > |Imaging Quality|0.633|0.590|
> > > |Overall Consistency|0.236|0.268|
> > > |Background Consistency|0.989|0.963|
> > > |Object Class|0.762|0.785|
> > > |Multiple Objects|0.545|0.488|
> > > |Color|0.872|0.889|
> > > |Spatial Relationship|0.700|0.471|
> > > |Scene|0.249|0.463|
> > > |Temporal Style|0.239|0.237|
> > > |Human Action|0.730|0.950|
> > > |Temporal Flickering|0.993|0.995|
> > > |Appearance Style|0.199|0.234|
> > > |Quality Score|0.835|0.806|
> > > |Semantic Score|0.657|0.706|
> > > |Total Score|0.800|0.786|
> > > |PSNR|28.12|21.33|

---

### Official Review · Reviewer_eRBx · 2025-10-29

**Soundness:** 3
**Presentation:** 3
**Contribution:** 3
**Rating:** 4
**Confidence:** 3

**Summary:**

This paper presents **ASTRAEA**, a token-level acceleration framework designed to mitigate the high computational costs of video diffusion transformers (vDiTs). The method introduces a lightweight token selection mechanism, a GPU-friendly sparse attention strategy, and an evolutionary search framework to find optimal token budgets across denoising steps.

**Strengths:**

- **Excellent Scalability**: A key strength of this work is its demonstrated scalability. The framework achieves strong performance scaling across multiple GPUs, showing up to 13.2x speedup on 8 GPUs, which highlights its practical utility for large-scale inference.

- **Good Performance Gains**: The proposed method delivers acceptable and noteworthy performance gains, achieving up to 2.4x inference speedup on a single GPU while maintaining high video quality (e.g., <0.5% VBench loss).

- **Clear Presentation**: The paper is well-written and logically organized. The core concepts, including the token selection mechanism and the sparse attention strategy, are presented clearly, making the work easy to follow.

**Weaknesses:**

- **Evolutionary Algorithm (EA) Search Cost**: The EA search for finding the optimal token distribution is computationally expensive, with an average search time of 82 GPU hours and some models taking up to 139 hours. While the authors rightly point out this is an offline cost, this is a significant practical hurdle. Further work could be done to optimize this search or analyze potential inefficiencies in the EA process.
- **Lack of Theoretical Justification for EA**: The paper's justification for using an EA is primarily empirical. While the experiments show great results, the paper would be stronger with a more detailed explanation of why EA is well-suited for this specific problem over other search methods. The current justification—that other methods like NAS have "substantial search times"—could be expanded upon with a brief analysis of the search space itself.
- **Missed Structural Optimizations**: The token selection framework appears to treat all tokens uniformly. It does not seem to leverage the inherent structures of video models, such as spatial-temporal sparsity. Furthermore, it does not analyze or exploit the different roles of text tokens versus spatio-temporal video tokens, which represents a missed opportunity for a more granular and potentially more effective optimization.
- **No Hyperparameter Sensitivity Analysis**: The core token selection metric in Equation 4 relies on two key hyperparameters, $w_{\alpha}$ and $w_{\beta}$, to weigh token significance and the non-selection penalty. The paper does not provide a sensitivity analysis for these hyperparameters. This makes it difficult to assess the robustness of the method and understand how critical fine-tuning these parameters is to achieving the reported results. The ablation study in Section 4.3 focuses on high-level design choices rather than these specific equation parameters.

**Questions:**

See weakness.

---

> ### Author Response · Authors · 2025-11-21
>
> ## Q1: EA search cost is high.
>
> EA might be time-costly at first glance. However, EA can achieve near-optimal results without any human-in-the-loop tuning. Although other training-free techniques lack an explicit training process, they all require laborious experiments to identify potential correlations/redundancies within models. Identifying those hyperparameters within their techniques is still time-consuming and has not been explicitly counted into their “human training” process. Thus, we believe EA simply automates this “human training” process and avoids suboptimal solutions in prior work.
>
> That said, we provide two techniques to accelerate EA. 1) EA can be easily parallelized by multiple GPUs. We can achieve 7.6x speedup with 8 GPUs.
> 2) EA can be further accelerated by dynamic programming. Since many EA candidates have the same prefix schedule (e.g., the first 20 timesteps being identical). We can evaluate those candidates together to avoid redundant computations. Overall, dynamic programming can further give 1.5x speedup.
>
> Lastly, the result from EA has great portability. I.e., results from one model can be used for others. Here is an example using the result from HunyuanVideo to test on Wan 2.1.
>
> |VBench Metric|Astraea 50%|Astraea 50% using Hunyuan|
> |-|-|-|
> |quality score|0.835|0.842|
> |semantic score|0.657| 0.620|
> |total score|0.800|0.798|
> |PSNR|28.10|27.75|
>
> ## Q2: Lack of theoretical justification for using EA.
>
> We expand our justification below.
>
> First, it is crucial to distinguish our task from Neural Architecture Search (NAS) or Network Pruning. [NAS](https://arxiv.org/abs/1808.05377) is defined by three components: Search Space (the model architecture), Search Algorithm (e.g., EA), and Evaluation Strategy. The NAS search space primarily searches the network architecture rather than the token budgets. On the other hand, [network pruning](https://arxiv.org/pdf/2308.06767) aims to remove redundant architectural components and often requires fine-tuning to avoid performance degradation. Both NAS and Network Pruning are more computationally intensive than ours. For instance, [NASNet](https://arxiv.org/pdf/1707.07012) required approximately 2,000 GPU hours to complete its architecture search, while [Diff-Pruning](https://arxiv.org/pdf/2305.10924) demanded up to 20% of the original model training expenditure.
>
> Our work, in contrast, focuses on token selection across timesteps—a task that pertains to the model's input rather than its architecture. As a result, NAS and Network Pruning cannot directly apply to our problem. The token selection formulation aligns well with the process of EA. So we use EA instead.
>
> ## Q3: Missed structural optimizations; your technique does not leverage the spatio-temporal sparsity in video tokens.
>
> True. Similar to all techniques (e.g., ToCa, PAB, Delta-DIT, TeaCache, DuCa, etc.) that ONLY leverage the temporal redundancies across timesteps. All these studies DO NOT leverage the spatio-temporal correlations embedded in vDiTs. Because these are two orthogonal directions to accelerate vDiTs: one is to reduce the computation redundancies due to the large timesteps, the other is to reduce the computation within a single timestep or attention block. All aforementioned studies focus on the former. The latter has been extensively studied by SVG, radicalAttention, MInference, etc.
>
> Nevertheless, our technique can be easily integrated with the techniques in the latter category. Here, we show that combining with SVG can achieve an additional 1.67x speedup with similar generative quality as SVG.
>
> Wan2.1 1.3B
> ||Vbench Score|PSNR|SSIM|LIPIS|Speedup|
> |-|-|-|-|-|-|
> |org|0.815|-|-|-|1.00|
> |Astraea 50%|0.811|25.67|0.884|0.071|1.91|
> |SVG|0.763|17.82|0.654|0.337|1.42|
> |SVG+Astraea 50%|0.770|18.50|0.677|0.309|2.37|
>
> From the table above, our method achieves better quality than SVG. Because SVG purely skips the insignificant attention scores in self-attention. However, part of SVG+Astraea still reuses the token results from the previous timestep. The result shows that reusing is more effective than purely skipping the computation. Thus, our combined variant can better preserve quality.
>
> ## Q4: Hyperparameter sensitivity for Equation (4).
>
> We performed a hyperparameter sensitivity study for Equation (4)’s parameters: $W_{\alpha}$: significance term; $W_{\beta}$: penalty term for non-selected tokens.
>
> We fixed $W_{\alpha} = 1$ and varied $W_{\beta}$. Below, we show the PSNR result on Wan2.1:
>
> |$W_\beta$|0.1|0.3|0.5|0.7|0.9|1.5|
> |-|-|-|-|-|-|-|
> |PSNR|25.63|25.65|25.67|25.64|25.60|25.46|
>
> The results show: $W_{\alpha}=1,W_{\beta}=0.5$ is the optimal setting. The overall variation is extremely small (<0.2 PSNR), indicating that our method is robust to these hyperparameters.
> Also, we would like to emphasize that this penalty term is adapted directly from ToCa, and our sensitivity trend matches the trend reported in ToCa. Thus, we claim no contribution to this term, as mentioned in the paper.

---

### Official Review · Reviewer_vjj7 · 2025-11-01

**Soundness:** 3
**Presentation:** 3
**Contribution:** 3
**Rating:** 6
**Confidence:** 2

**Summary:**

This paper presents  a framework for accelerating vDiTs via token selection. The method leverages the multi-timestep nature of diffusion models for token selection, resulting to a GPU-efficient sparse attention strategy, achieving linear inference speedup with minimal quality degradation.

**Strengths:**

- The idea is novel and conceptually makes sense. It cleverly leverages the multi-timestep nature of diffusion models for token selection.
- The method is supported by solid experiments, and the resulting performance is highly impressive.

**Weaknesses:**

Since the paper claims to outperform native sparse attention in line 233 to line 241, would it be possible to include a performance comparison with state-of-the-art methods, such as SVG2 [1]?

[1] Sparse VideoGen2: Accelerate Video Generation with Sparse Attention via Semantic-Aware Permutation

**Questions:**

- Line 203, 204: `We use the LSE score in the previous timestep because LSE scores do not vary across timestep`. What is the basis for this claim?
- Why does Table 1, lines 379–380 say `on two vDiT models` while in fact there are three models?

---

> ### Author Response · Authors · 2025-11-21
>
> Thanks for your valuable feedback!
>
> ## Q1: Comparison with SOTA methods such as SVG2.
>
> Sure, we include the result from SVG2. Here, we show the results on two best-performing vDiT models, HunyuanVideo and Wan2.1, using their officially released inference configurations. The results are shown below:
>
> HunyuanVideo
> |Metric|Original|SVG2|Astraea70%|Astraea50%|Astraea40%|
> |-|-|-|-|-|-|
> |Quality Score|0.837|0.829|0.844|0.838|0.835|
> |Semantic Score|0.666|0.723|0.648|0.659|0.650|
> |Total Score|0.803|0.808|0.804|0.802|0.798|
> |PSNR|–|21.52|33.62|28.71|27.61|
> |Speedup|1.00|1.53|1.39|1.93|2.38|
>
> Wan2.1 1.3B 33frame
> |Metric|Original|SVG2|Astraea70%|Astraea50%|Astraea40%|
> |-|-|-|-|-|-|
> |Quality Score|0.843|0.839|0.842|0.841|0.839|
> |Semantic Score|0.699|0.617|0.697|0.690|0.687|
> |Total Score|0.815|0.795|0.813|0.811|0.808|
> |PSNR|–|22.28|30.83|25.67|23.77|
> |Speedup|1.00|1.13|1.39|1.91|2.35|
>
> Wan2.1 1.3B 65frame
> |Metric|Original|SVG2|Astraea70%|Astraea50%|Astraea40%|
> |-|-|-|-|-|-|
> |QualityScore|0.837|0.830|0.837|0.835|0.834|
> |SemanticScore|0.666|0.640|0.661|0.657|0.652|
> |TotalScore|0.808|0.792|0.802|0.800|0.798|
> |PSNR|–|22.95|33.00|28.12|26.98|
> |Speedup|1.00|1.32|1.37|1.91|2.38|
>
> Overall, our method consistently outperforms SVG2 with better quality consistency. This shows that Astraea can leverage model temporal consistency to dynamically allocate sparse tokens more effectively than fixed semantic-based sparse patterns.
>
> ## Q2: Basis for the claim that LSE scores do not vary across timesteps.
>
> To support this claim, we analyzed attention LSE statistics across timesteps. Specifically, we show the LSE value of the same attention block across consecutive diffusion timesteps.
>
> First, we calculated the cosine similarity of the LSE of the attention for each block in adjacent timesteps during the inference process of the Wan2.1 model. The results are as follows.
>
> |Mean|Variance|Median|
> |-|-|-|
> |99.1%|0.00736% |0.858%|
>
> Second, we prove the case where the difference in latent x is less than $\delta$ (which is small for two adjacent timesteps), and then, the difference in LSE is also small (only a constant multiple of $\delta$).
>
> Let the attention logits be,
>
> $a_i = \frac{1}{\sqrt{d}}\, q^\top k_i,$
>
> computed from queries $q = W_Q x$ and keys $k_i = W_K x_i$.
> Assume the inputs at adjacent timesteps satisfy $\|x_t - x_{t-1}\| < \delta$. Then,
>
> $\|q_t - q_{t-1}\| \le \|W_Q\|\,\delta, \qquad \|k_{i,t} - k_{i,t-1}\| \le \|W_K\|\,\delta.$
>
> For the change in logits:
>
> $ a_{i,t} - a_{i,t-1} = \frac{1}{\sqrt d} \left[(q_t-q_{t-1})^\top k_{i,t} + q_{t-1}^\top (k_{i,t}-k_{i,t-1})\right],$
>
> and therefore
>
> $ \|a_{i,t}-a_{i,t-1}\| \le \frac{1}{\sqrt d} \left( \|W_Q\|\,\|k_{i,t}\| + \|W_K\|\,\|q_{t-1}\| \right)\delta.$
>
> Applying the 1-Lipschitz property of LSE,
>
> $\|LSE_t - LSE_{t-1}\| \le \max_{i} |a_{i,t}-a_{i,t-1}| \le  \frac{1}{\sqrt{d}} ( \|\|W_Q\|\|\max_{i}\| \|k_i\|\| + \|\|W_K\|\|\max_i \|\|q\|\| )\delta.$
>
> Hence, small changes in the block inputs at adjacent diffusion timesteps lead to proportionally small changes in the softmax log-sum-exp values of the attention logits.
>
> Overall, LSE values exhibit high temporal consistency, with only small variations across timesteps. This is also the assumption of many token-reusing techniques, such as ToCa. They reuse the attention scores from the previous timestep. The basic assumption from ToCa is that attention scores will not change significantly across timesteps.
>
>
> ## Q3: Why does Table 1 say two vDiT models when there are actually three?
>
> We thank the reviewer for catching this typo. We conduct experiments with three models. We will update the table and ensure consistency throughout the paper.

---

### Official Review · Reviewer_nMzk · 2025-11-01

**Soundness:** 3
**Presentation:** 3
**Contribution:** 2
**Rating:** 6
**Confidence:** 2

**Summary:**

This paper proposes ASTRAEA, a token-wise acceleration framework for video diffusion transformers (vDiTs). Unlike prior acceleration methods that rely on step reduction or blocking caching, ASTRAEA introduces:

1. A lightweight token selection mechanism that dynamically identifies important tokens at each denoising step with negligible overhead.
2. A GPU-friendly sparse attention strategy that selectively computes queries while retaining full keys/values, ensuring correctness and parallelizability.
3. An evolutionary search framework to automatically allocate token budgets across timesteps, avoiding manual hyperparameter tuning.

**Strengths:**

1. Operating at the token level, it offers finer granularity than step- or block-level methods, addressing a previously underexplored dimension.
2. Evolutionary algorithm effectively allocates token budgets across timesteps, reducing reliance on heuristics.
3. Clear evidence of multi-GPU efficiency, which is crucial for industrial deployment.

**Weaknesses:**

1. Evolutionary search may still be computationally expensive, limiting practicality in large-scale or frequently updated deployments.
2. Generality of prompts: Search is conducted on a small set of prompts; broader validation on diverse datasets would strengthen claims of generalization.

**Questions:**

1. Generalization: If the token budget allocation is searched on one dataset or prompt set, how well does it transfer to unseen prompts or domains?
2. Integration with retraining methods: Could ASTRAEA be combined with distillation-based step reduction to achieve further acceleration?

---

> ### Author Response · Authors · 2025-11-21
>
> Thanks for your valuable feedback!
>
> ## Q1: Evolutionary search (EA) is still computationally expensive.
>
> EA might be time-costly at first glance. However, EA can achieve near-optimal results without any human-in-the-loop tuning. Although other training-free techniques do not have an explicit training process, they all require laborious experiments to identify potential correlations/redundancies within models. We believe identifying those hyperparameters within their techniques is still time-consuming and has not been explicitly counted into their “human training” process. Thus, we believe EA simply automates this “human training” process and avoids suboptimal solutions in prior work.
>
> That said, we provide two techniques to accelerate EA:
> 1) EA can be easily parallelized by multiple GPUs. Different candidates can be distributed to different GPUs. Doing so, we can achieve 7.6x speedup with 8 GPUs. Thus, EA can be done in one day.
> 2) EA can be further accelerated by dynamic programming. Since many EA candidates have the same prefix schedule (e.g., the first 20 timesteps being identical). We can evaluate those candidates together by saving the intermediate results to avoid redundant computations. Overall, we find that dynamic programming can further provide 1.5x speedup.
>
> Combining both techniques, we can achieve 11.4x speedup.
>
> Lastly, we find that the result from EA has great portability. I.e., results from one model can be used for others. Here, we provide an example using the result from HunyuanVideo to test on Wan 2.1. (They both have 50 timesteps)
>
> | VBench Metric| Astraea 50% | Astraea 50% using HunyuanVideo EA result |
> |-|-|-|
> |subject consistency|0.959|0.958|
> |motion smoothness|0.983|0.984|
> |dynamic degree|0.625|0.643|
> |aesthetic quality|0.616|0.647|
> |imaging quality|0.633|0.642|
> |overall consistency|0.236|0.223|
> |background consistency| 0.989| 0.986|
> |object class | 0.762| 0.762|
> |multiple objects| 0.545| 0.465|
> |color|0.872|0.745|
> |spatial relationship|0.700|0.718|
> |scene|0.249| 0.235|
> |temporal style | 0.239| 0.250|
> |human action | 0.730|0.600|
> |temporal flickering | 0.993| 0.991|
> | appearance style| 0.199| 0.203|
> | quality score| 0.835|0.842|
> | semantic score | 0.657| 0.620|
> | total score| 0.800|0.798|
> |PSNR| 28.10 | 27.75|
>
> In this way, EA search time can be amortized across multiple vDiT models.
>
> ## Q2: Generality of prompts and transfer to unseen prompts.
>
> Yes, our method is explicitly designed to test cross-prompt generalization. Here, EA is conducted with ONLY 4 prompts for all models (Hunyuan, Wan, Opensora). The final evaluation is performed on VBench, which contains >900 prompts covering various domains and styles, and our technique achieves the best performance across all vDiT models.
>
> In Fig. 5, we show the model redundancies across timesteps with 8 randomly picked prompts (we remove one specific timestep out of the entire denoising process and evaluate its MSE loss against the original baseline). For all vDiT models, the results show a similar trend across 8 prompts. Thus, we believe our technique is largely generalizable.
>
> Also, we leverage nothing but the model redundancy across timesteps, similar to PAB, ToCa, Delta-Cache, etc. All prior studies use static reusing strategies. This means that the model redundancy is an inherent feature in vDiTs. We simply pick some prompts and automate this “redundancy-elimination” process.
>
> Results from Q1 can also verify the generality of our technique. The token reusing policy is also portable across different models.
>
> ## Q3: Can Astraea be combined with distillation-based step reduction?
>
> Potentially yes. We combined ASTRAEA with LightX2V, a distilled 4-step variant of Wan2.1-14B (step-reduction via distillation). We applied our EA-based token allocation search on top of LightX2V’s 4-step inference flow. The overall result achieves further acceleration with a marginal quality compromise. Here is the result:
>
> |VBench Metric| Lightx2v | Astraea 70%|
> |-|-|-|
> |subject consistency|0.971|0.971|
> |motion smoothness|0.980|0.986|
> |dynamic degree|0.857|0.857|
> |aesthetic quality|0.647|0.621|
> |imaging quality|0.688|0.696|
> |overall consistency |0.242| 0.244|
> |background consistency | 0.936 | 0.934|
> |object class| 0.988| 1.000 |
> |multiple objects| 0.656 | 0.707|
> |color| 0.921| 0.907|
> |spatial relationship| 0.867 | 0.772|
> |scene| 0.423| 0.445|
> |temporal style | 0.237| 0.234|
> |human action | 0.850| 0.750|
> |temporal flickering | 0.952| 0.971|
> |appearance style|0.208|0.209|
> |quality score|0.839|0.847|
> |semantic score|0.760|0.747 |
> |total score|0.823|0.827|
> |speedup|1.00|1.14|
>
> However, both Astraea and any distillation methods leverage the model redundancies. This means that Astraea would have a little performance gain over aggressively distilled models. Nevertheless, we believe Astraea is more lightweight. E.g., distilling a Wan2.1 1.3B requires 2000 GPU hours, which is way much higher than Astraea.

---

### Author Response · Authors · 2025-11-23

We thank the reviewers for their constructive feedback!

During the rebuttal phase, we made the following major changes to improve the manuscript. Additionally, we respond to individual reviewer questions that might not be mentioned below. Please refer to the individual responses for more details. The major modifications in the revision are summarized below:

1. **Clarification on LSE usage (Line 204–207)**: We added an explanation that the Log-Sum-Exp (LSE) of the same block varies minimally across adjacent timesteps, which allows us to use the previous timestep’s LSE as an effective approximation of token importance.

2. **Justification for using Evolutionary Algorithms (EA) (Line 330–338)**: We expanded the discussion to explain why EA is more suitable for our token-selection search space compared to NAS or network pruning.

3. **Typo corrections**: Several typos pointed out by reviewers were fixed throughout the paper.

4. **Updated Table 1 with SVG2 baseline**: We added SVG2 as an additional baseline for both HunyuanVideo and Wan to strengthen the comparison.

5. **Added breakdown for three vDiT models (Fig. 7 and Line 455–460)**: We included runtime breakdowns for three vDiT architectures to demonstrate that our token-selection mechanism introduces only minimal overhead.

6. **Added sensitivity study for Eqn. (4) hyperparameters (Fig. 10 and Line 505–517)**: We reported a sensitivity analysis of the hyperparameters in the token-score formulation.

7. **Expanded EA search-time analysis in Supplementary (Line 969–983)**: We added a discussion on the EA search time and described potential strategies to accelerate the search process.

---

### Comment · Area_Chair_9tBW · 2025-11-27

Dear Reviewers,

Thank you to those who have already reviewed the author responses and updated your comments - much appreciated！

For the remaining reviews, this is a gentle reminder to please take a moment to check the rebuttal and indicate whether the authors’ responses address your concerns. Even a brief note is very helpful for the meta-review and final decision process.

Thanks again for all your time and contributions to ensuring a fair and high-quality review cycle.

Best,
AC

---

### Author Response · Authors · 2025-12-02
**Rebuttal Summary to New AC**

Dear AC,

We sincerely thank your efforts and all reviewers for their valuable feedback! We summarize our answers to all questions and concerns from reviewers below:

## Q1: EA is time-consuming. (Review nMzk, eRBx, hZdk)

**Short answer**: This is the biggest concern raised by most reviewers. While search time is less critical in our setting (the EA search is done offline and can benefit billions of users all over the world), we have further reduced the EA search time from **82 GPU hours** to **1.4 amortized GPU hours**. We believe this substantially addresses the reviewers’ concerns.

**Long answer**: The detailed optimization process is described below.

First, we have to clarify an important point: the reported search time of 82 GPU hours refers to the total search time, i.e., we only require roughly **10 hours** on a cluster of 8 A100 GPUs.

Second, with the additional memoization technique, we can further reduce the search time to **7.2 hours** (1.5x speedup). Since many EA candidates have the same prefix schedule (e.g., the first 20 timesteps being identical). We can evaluate those candidates together by saving the intermediate results to avoid redundant computations. Thus, we can use the classic memoization technique to further reduce the search cost.

Third, we show that one EA result from a single model is portable across models, i.e., this EA result can be used on other vDiT models. Here, we show that a schedule searched on HunyuanVideo can transfer to Wan2.1 and OpenSora. Note that the total timestep of OpenSora is 30. Thus, we linearly downsample the Hunyuan search result from 50 to 30 timesteps and apply it to OpenSora. This way, the search time is further amortized to 1.4 hours. **We DO NOT think 1.4 hours is “time-consuming” at all**. The detailed quality comparison is shown below.

Lastly, we want to emphasize that one search result can be reused across millions of inference services. Given our collaboration with one of the leading cloud service providers in the world, ``half a day’’ search time is negligible, especially considering the substantial gains in quality and performance of our approach over prior work.

Also, I want to emphasize that **all prior work is NOT free of cost**. Although they do not have an explicit search process, they all require laborious experiments to identify potential correlations/redundancies within each vDiT model. Identifying those hyperparameters used in their techniques is still a time-consuming process and has not been explicitly counted into their  "human search" process. Our EA algorithm simply automates this "human search’" process in a more elegant way and avoids suboptimal solutions in prior work.

The overall results are shown below (we only show the results from 4-second videos):
|Metric|OpenSora Astraea 50%|OpenSora Astraea 50% using HunyuanVideo EA result|Wan Astraea 50%|Wan Astraea 50% using HunyuanVideo EA result|
|-|-|-|-|-|
|Subject Consistency|0.946|0.936|0.959|0.958|
|Motion Smoothness|0.987|0.978|0.983|0.984|
|Dynamic Degree|0.472|0.643|0.625|0.643|
|Aesthetic Quality|0.548|0.552|0.616|0.647|
|Imaging Quality|0.575|0.559|0.633|0.642|
|Overall Consistency|0.272|0.264|0.236|0.223|
|Background Consistency|0.972|0.957|0.989|0.986|
|Object Class|0.790|0.785|0.762|0.762|
|Multiple Objects|0.470|0.504|0.545|0.465|
|Color|0.822|0.852|0.872|0.745|
|Spatial Relationship|0.604|0.409|0.700|0.718|
|Scene|0.422|0.452|0.249|0.235|
|Temporal Style|0.239|0.235|0.239|0.250|
|Human Action|0.870|0.800|0.730|0.600|
|Temporal Flickering|0.995|0.995|0.993|0.991|
|Appearance Style|0.237|0.233|0.199|0.203|
|Quality Score|0.801|0.803|0.835|0.842|
|Semantic Score|0.700|0.676|0.657|0.620|
|Total Score|0.781|0.778|0.800|0.798|
|PSNR|28.51|24.69|28.12|27.75|

Overall, using a schedule from other models would decrease the generative quality. However, it is still significantly better than prior work.

---

> ### Author Response · Authors · 2025-12-02
>
> ## Q2: Hyperparameter sensitivity for Equation 4. (Reviewer eRBx, hZdk)
>
> **Short answer**: We update the result in Fig. 10 and Lines 505–517.
>
> **Long answer**: We performed a hyperparameter sensitivity study for Equation (4)’s parameters: $W_{\alpha}$: significance term; $W_{\beta}$: penalty term for non-selected tokens.
>
> We fixed $W_{\alpha} = 1$ and varied $W_{\beta}$. Below, we show the PSNR result on Wan2.1:
>
> |$W_\beta$|0.1|0.3|0.5|0.7|0.9|1.5|
> |-|-|-|-|-|-|-|
> |PSNR|25.63|25.65|25.67|25.64|25.60|25.46|
>
> The results show: $W_{\alpha}=1,W_{\beta}=0.5$ is the optimal setting. The overall variation is extremely small (<0.2 PSNR), indicating that our method is robust to these hyperparameters.
> Also, we would like to emphasize that this penalty term is adapted directly from ToCa, and our sensitivity trend matches the trend reported in ToCa. Thus, we claim no contribution to this term, as mentioned in the paper.
>
>
> ## Q3: Generality of prompts and transfer to unseen prompts. (Reviewer nMzk)
>
> **Short answer**: Our method is highly robust in generalization. We explicitly search using a small set of 4 prompts and test over 900 prompts.
>
> **Long answer**: Yes, our method is explicitly designed to test cross-prompt generalization. Here, EA is conducted with ONLY 4 prompts for all models (Hunyuan, Wan, Opensora). The final evaluation is performed on VBench, which contains >900 prompts covering various domains and styles, and our technique achieves the best performance across all vDiT models.
>
> In Fig. 5, we show the model redundancies across timesteps with 8 randomly picked prompts (we remove one specific timestep out of the entire denoising process and evaluate its MSE loss against the original baseline). For all vDiT models, the results show a similar trend across 8 prompts. Thus, we believe our technique is largely generalizable.
>
> Also, we leverage nothing but the model redundancy across timesteps, similar to PAB, ToCa, Delta-Cache, etc. All prior studies use static reusing strategies. This means that the model redundancy is an inherent feature in vDiTs. We simply pick some prompts and automate this “redundancy-elimination” process.
>
> Results from Q1 can also verify the generality of our technique. The token reusing policy is also portable across different models.
>
> ## Q4: Can Astraea be combined with distillation-based step reduction? (Reviewer nMzk)
>
> **Short answer**: Yes. See the result below.
>
> **Long answer**: We combined ASTRAEA with LightX2V, a distilled 4-step variant of Wan2.1-14B (step-reduction via distillation). We applied our EA-based token allocation search on top of LightX2V’s 4-step inference flow. The overall result achieves further acceleration with a marginal quality compromise. Here is the result:
>
> |VBench Metric| Lightx2v | Astraea 70%|
> |-|-|-|
> |subject consistency|0.971|0.971|
> |motion smoothness|0.980|0.986|
> |dynamic degree|0.857|0.857|
> |aesthetic quality|0.647|0.621|
> |imaging quality|0.688|0.696|
> |overall consistency |0.242| 0.244|
> |background consistency | 0.936 | 0.934|
> |object class| 0.988| 1.000 |
> |multiple objects| 0.656 | 0.707|
> |color| 0.921| 0.907|
> |spatial relationship| 0.867 | 0.772|
> |scene| 0.423| 0.445|
> |temporal style | 0.237| 0.234|
> |human action | 0.850| 0.750|
> |temporal flickering | 0.952| 0.971|
> |appearance style|0.208|0.209|
> |quality score|0.839|0.847|
> |semantic score|0.760|0.747 |
> |total score|0.823|0.827|
> |speedup|1.00|1.14|
>
> However, both Astraea and any distillation methods leverage the model redundancies. This means that Astraea would have a little performance gain over aggressively distilled models. Nevertheless, we believe Astraea is more lightweight. E.g., distilling a Wan2.1 1.3B requires 2000 GPU hours, which is way much higher than Astraea.

---

> > ### Author Response · Authors · 2025-12-02
> >
> > ## Q5: Comparison with SOTA methods such as SVG2. (Reviewer vjj7)
> >
> > **Short answer**: We include the result from SVG2 in Table 1, and it is worse than our method.
> >
> > **Long answer**: We show the results on two best-performing vDiT models, HunyuanVideo and Wan2.1, using their officially released inference configurations. The results are shown below:
> >
> > HunyuanVideo
> > |Metric|Original|SVG2|Astraea70%|Astraea50%|Astraea40%|
> > |-|-|-|-|-|-|
> > |Quality Score|0.837|0.829|0.844|0.838|0.835|
> > |Semantic Score|0.666|0.723|0.648|0.659|0.650|
> > |Total Score|0.803|0.808|0.804|0.802|0.798|
> > |PSNR|–|21.52|33.62|28.71|27.61|
> > |Speedup|1.00|1.53|1.39|1.93|2.38|
> >
> > Wan2.1 1.3B 33frame
> > |Metric|Original|SVG2|Astraea70%|Astraea50%|Astraea40%|
> > |-|-|-|-|-|-|
> > |Quality Score|0.843|0.839|0.842|0.841|0.839|
> > |Semantic Score|0.699|0.617|0.697|0.690|0.687|
> > |Total Score|0.815|0.795|0.813|0.811|0.808|
> > |PSNR|–|22.28|30.83|25.67|23.77|
> > |Speedup|1.00|1.13|1.39|1.91|2.35|
> >
> > Wan2.1 1.3B 65frame
> > |Metric|Original|SVG2|Astraea70%|Astraea50%|Astraea40%|
> > |-|-|-|-|-|-|
> > |QualityScore|0.837|0.830|0.837|0.835|0.834|
> > |SemanticScore|0.666|0.640|0.661|0.657|0.652|
> > |TotalScore|0.808|0.792|0.802|0.800|0.798|
> > |PSNR|–|22.95|33.00|28.12|26.98|
> > |Speedup|1.00|1.32|1.37|1.91|2.38|
> >
> > Overall, our method consistently outperforms SVG2 with better quality consistency. This shows that Astraea can leverage model temporal consistency to dynamically allocate sparse tokens more effectively than fixed semantic-based sparse patterns.
> >
> >
> > ## Q6: Basis for the claim that LSE scores do not vary across timesteps. (Reviewer vjj7)
> >
> > **Short answer**: We provide both LSE statistics across timesteps and the rigorous proof to show that LSE scores do not vary across timesteps.
> >
> > **Long answer**: To support this claim, we analyzed attention LSE statistics across timesteps. Specifically, we show the LSE value of the same attention block across consecutive diffusion timesteps.
> >
> > First, we calculated the cosine similarity of the LSE of the attention for each block in adjacent timesteps during the inference process of the Wan2.1 model. The results are as follows.
> >
> > |Mean|Variance|Median|
> > |-|-|-|
> > |99.1%|0.00736% |0.858%|
> >
> > Second, we prove the case where the difference in latent x is less than $\delta$ (which is small for two adjacent timesteps), and then, the difference in LSE is also small (only a constant multiple of $\delta$).
> >
> > Let the attention logits be,
> >
> > $a_i = \frac{1}{\sqrt{d}}\, q^\top k_i,$
> >
> > computed from queries $q = W_Q x$ and keys $k_i = W_K x_i$.
> > Assume the inputs at adjacent timesteps satisfy $\|x_t - x_{t-1}\| < \delta$. Then,
> >
> > $\|q_t - q_{t-1}\| \le \|W_Q\|\,\delta, \qquad \|k_{i,t} - k_{i,t-1}\| \le \|W_K\|\,\delta.$
> >
> > For the change in logits:
> >
> > $ a_{i,t} - a_{i,t-1} = \frac{1}{\sqrt d} \left[(q_t-q_{t-1})^\top k_{i,t} + q_{t-1}^\top (k_{i,t}-k_{i,t-1})\right],$
> >
> > and therefore
> >
> > $ \|a_{i,t}-a_{i,t-1}\| \le \frac{1}{\sqrt d} \left( \|W_Q\|\,\|k_{i,t}\| + \|W_K\|\,\|q_{t-1}\| \right)\delta.$
> >
> > Applying the 1-Lipschitz property of LSE,
> >
> > $\|LSE_t - LSE_{t-1}\| \le \max_{i} |a_{i,t}-a_{i,t-1}| \le  \frac{1}{\sqrt{d}} ( \|\|W_Q\|\|\max_{i}\| \|k_i\|\| + \|\|W_K\|\|\max_i \|\|q\|\| )\delta.$
> >
> > Hence, small changes in the block inputs at adjacent diffusion timesteps lead to proportionally small changes in the softmax log-sum-exp values of the attention logits.
> >
> > Overall, LSE values exhibit high temporal consistency, with only small variations across timesteps. This is also the assumption of many token-reusing techniques, such as ToCa. They reuse the attention scores from the previous timestep. The basic assumption from ToCa is that attention scores will not change significantly across timesteps.
> >
> >
> > ## Q7: Why does Table 1 say two vDiT models when there are actually three?  (Reviewer vjj7)
> >
> > We thank the reviewer for catching this typo. We conduct experiments with three models. We will update the table and ensure consistency throughout the paper.

---

> > > ### Author Response · Authors · 2025-12-02
> > >
> > > ## Q8: Lack of theoretical justification for using EA. (Reviewer eRBx)
> > >
> > > First, it is crucial to distinguish our task from Neural Architecture Search (NAS) or Network Pruning. [NAS](https://arxiv.org/abs/1808.05377) is defined by three components: Search Space (the model architecture), Search Algorithm (e.g., EA), and Evaluation Strategy. The NAS search space primarily searches the network architecture rather than the token budgets. On the other hand, [network pruning](https://arxiv.org/pdf/2308.06767) aims to remove redundant architectural components and often requires fine-tuning to avoid performance degradation. Both NAS and Network Pruning are more computationally intensive than ours. For instance, [NASNet](https://arxiv.org/pdf/1707.07012) required approximately 2,000 GPU hours to complete its architecture search, while [Diff-Pruning](https://arxiv.org/pdf/2305.10924) demanded up to 20% of the original model training expenditure.
> > >
> > > Our work, in contrast, focuses on token selection across timesteps—a task that pertains to the model's input rather than its architecture. As a result, NAS and Network Pruning cannot directly apply to our problem. The token selection formulation aligns well with the process of EA. So we use EA instead.
> > >
> > > ## Q9: Missed structural optimizations; your technique does not leverage the spatio-temporal sparsity in video tokens. (Reviewer eRBx)
> > >
> > > **Short answer**: True. Similar to all techniques (e.g., ToCa, PAB, Delta-DIT, TeaCache, DuCa, etc.), we ONLY leverage the temporal redundancies across timesteps. However, our technique can integrate with the methods that leverage the spatio-temporal sparsity and achieve an additional 1.67x speedup with the same quality as SVG.
> > >
> > > **Long answer**: True. Similar to all techniques (e.g., ToCa, PAB, Delta-DIT, TeaCache, DuCa, etc.), we ONLY leverage the temporal redundancies across timesteps. All these studies DO NOT leverage the spatio-temporal correlations embedded in vDiTs. Because these are two orthogonal directions to accelerate vDiTs: one is to reduce the computation redundancies due to the large timesteps, the other is to reduce the computation within a single timestep or attention block. All aforementioned studies focus on the former. The latter has been extensively studied by SVG, radicalAttention, MInference, etc.
> > >
> > > Nevertheless, our technique can be easily integrated with the techniques in the latter category. Here, we show that combining with SVG can achieve an additional 1.67x speedup with similar generative quality as SVG.
> > >
> > > Wan2.1 1.3B
> > > ||Vbench Score|PSNR|SSIM|LIPIS|Speedup|
> > > |-|-|-|-|-|-|
> > > |org|0.815|-|-|-|1.00|
> > > |Astraea 50%|0.811|25.67|0.884|0.071|1.91|
> > > |SVG|0.763|17.82|0.654|0.337|1.42|
> > > |SVG+Astraea 50%|0.770|18.50|0.677|0.309|2.37|
> > >
> > > From the table above, our method achieves better quality than SVG. Because SVG purely skips the insignificant attention scores in self-attention. However, part of SVG+Astraea still reuses the token results from the previous timestep. The result shows that reusing is more effective than purely skipping the computation. Thus, our combined variant can better preserve quality.
> > >
> > > ## Q10: How to obtain the LSE score from FlashAttention? (Reviewer hZdk)
> > >
> > > **Short answer**: FlashAttention implementations (both v2 and v3) naturally provide the interface.
> > >
> > > **Long answer**: Current FlashAttention implementations (both v2 and v3) provide a dedicated interface to obtain the LSE score used internally for numerical stability. Specifically, you can obtain the LSE score via the official API by setting “return_attn_probs=True”. It will return the LSE score.
> > >
> > > Inside the FlashAttention kernel, it computes LSE in the fused kernel during the streaming softmax. So it requires ZERO extra computation and minimal memory overhead.

---

> > > > ### Author Response · Authors · 2025-12-02
> > > >
> > > > ## Q11: Sparse attention in Fig. 3 seems to add computation. Why claim lower complexity than QK sparsification? Sparse QK should be faster.  (Reviewer hZdk)
> > > >
> > > > **Short answer**: The reviewer misunderstood our claim. We claim “our sparse attention is more GPU-friendly,” although it “saves less computation than QK sparsification”.
> > > >
> > > > **Long answer**: Nowhere in Fig. 3 or throughout Section 3.2, we state or indicate “our sparse attention computation has lower complexity than QK sparsification.” We only claim that “our Q-only sparse attention is more GPU-friendly than QK sparsification” and “our approach saves less computation than naive sparse attention (QK sparsification)”.
> > > >
> > > > There are three reasons that our attention computation is more GPU-friendly (as we already stated in the paper):
> > > > 1. It does not require storing the entire attention scores from the previous timestep to fill the missing elements in the attention map. Storing the entire attention scores almost doubles the GPU memory compared to the baseline (see Table 1).
> > > > 2. All GPUs adopt a SIMT (single-instruction-multiple-thread) execution fashion. Unstructured QK sparsity (like our case) will achieve almost no speedup compared to Q-only sparsity.
> > > > 3. Lastly, QK-sparsity requires filling in missing attention scores in the attention map. It would introduce irregular memory accesses, which would further slow down the performance. ToCa in Table 1 uses QK sparsification. It is much slower than ours.
> > > >
> > > > ## Q12: Sparse QK theoretically should be faster. Can a faster hardware implementation be achieved using a kernel designed like SVG?  (Reviewer hZdk)
> > > >
> > > > **Short answer**: Incorrect. Unstructured sparse QK is extremely difficult to accelerate efficiently on modern AI accelerators—including GPUs, Google TPUs, and recent unicore architectures such as Cerebras, Groq, and Sambanova.
> > > >
> > > > **Long answer**: Very unlikely. On modern GPUs, accelerating irregular, unstructured sparse attention computation is always an open question. Those sparse attention computations often have low GPU utilization. To improve the GPU utilization, people often transform those sparse attentions into somewhat dense attention computation using some offline shuffling strategies, like SVG. However, our method dynamically selects reused tokens at runtime. Those offline strategies won’t help improve the performance.
> > > >
> > > > Additionally, sparse QK requires filling in the missing scores in the attention map. The memory access overheads are non-trivial.
> > > >
> > > >
> > > > ## Q13: Missing performance analysis of individual components.  (Reviewer hZdk)
> > > >
> > > > **Short answer**: We provide the performance breakdown in Fig. 7 and Line 455–460.
> > > >
> > > > **Long answer**: We provide the performance breakdown of ASTRAEA-50% on the three models (Wan2.1 1.3B, HunyuanVideo, OpenSora-1.2).
> > > >
> > > > Wan2.1 1.3B 65frame 480p, unit: second.
> > > > |Module|token select|self attn|cross attn|mlp|other|total|
> > > > |-|-|-|-|-|-|-|
> > > > |Astraea 50%|1.50|55.32|8.21|14.92|3.39|83.35|
> > > > |org|0.00|105.72|15.12|28.03|6.16|155.03|
> > > >
> > > > HunyuanVideo 129frame 544x960, unit: second.
> > > > |Mode|token select|self attn|cross attn|mlp|other|total|
> > > > |-|-|-|-|-|-|-|
> > > > |Astraea 50%|13.23|454.29|—|73.77|94.58|635.87|
> > > > |org|0.00|906.09|—|146.57|174.00|1226.66|
> > > >
> > > > OpenSora-1.2 93frame 480p, unit: second.
> > > > |Mode|token select|self attn|cross attn|mlp|other|total|
> > > > |-|-|-|-|-|-|-|
> > > > |Astraea 50%|1.86|22.82|9.93|19.88|4.12|58.61|
> > > > |org|0.00|43.32|19.53|38.36|7.82|109.02|
> > > >
> > > > ## Q14: Statement in line 463 contradicts line 464.
> > > >
> > > > Yes, it is indeed a typo, and we fixed it.

---

### Meta-Review · Area_Chair_Xk38 · 2026-01-07

**Summary:**

One of the main concerns is the practicality and justification of the evolutionary search (EA). Multiple reviewers highlight that the EA-based scheduling incurs a non-trivial offline cost, and question how broadly the searched schedule generalizes beyond the limited prompt set used during search. Several ask for either stronger evidence of transfer (across prompts/domains/datasets) or lighter alternatives.

Another concern is the clarity and validation of key technical claims. Reviewers request clarification on (i) the claim that LSE scores are stable across timesteps and (ii) how LSE is obtained efficiently and memory-friendly in practice (especially when using kernels like FlashAttention). They also question the paper’s narrative around why selecting only Q is preferable to sparsifying QK (which seems theoretically faster), and ask for clearer explanations and component-level profiling.

At least one reviewer explicitly requests direct comparison to SVG2. Others ask for component-wise performance analysis, hyperparameter sensitivity for the token scoring weights.

**Reviewer Concerns:**

Concerns Likely Addressed in the Authors’ Response
- Analysis of efficiency of EA search. The authors reduce the time cost further via (i) memoization (to 7.2 hours) and (ii) cross-model transfer of the searched schedule, amortizing to 1.4 GPU hours. They also provide a cross-model transfer quality table to show the transferred schedule remains competitive.
- Comparison to SVG2. They add explicit SVG2 results and claim Astraea consistently outperforms SVG2.
- How to obtain LSE from FlashAttention. They assert FlashAttention exposes an interface to return LSE and that LSE is computed inside the fused streaming softmax (thus negligible extra compute).
- Missing component-wise performance analysis. They provide a module breakdown for three models, directly answering “where the speedup comes from” and whether token selection overhead is significant.
- Combination with distillation-based step reduction. They provide a combined result with LightX2V and report modest additional speedup.

Concerns Partially Addressed
- Basis for the claim that LSE scores do not vary across timesteps. The response includes empirical statistics and theoretical analysis. This is a reasonable two-pronged justification, but it remains conditional (it relies on the assumption that adjacent latent differences are small).
- Prompt generalization. They argue EA searches with only 4 prompts but evaluates on 900+ VBench prompts and show similar redundancy trends across multiple prompts. This suggests generalization, but a more direct transfer ablations (e.g., schedules searched on prompt set A evaluated on prompt set B) could benefit.
- Why EA specifically. They distinguish their problem from NAS/pruning and say token selection matches EA well. However, the response is more “why NAS/pruning don’t apply” than “why EA is preferable to other optimization strategies” (e.g., greedy, Bayesian optimization, differentiable relaxation). This likely reduces but may not fully eliminate the concern.
- Not leveraging spatio-temporal sparsity within tokens. The authors essentially concede the point and reposition it as orthogonal, then show Astraea + SVG integration results. This mitigates the critique by showing complementarity, but it does not transform the missing direction into a contribution, so the concern is only partially relieved.

Concerns Still Likely Outstanding
- FlashAttention API precision / implementation exactness. The authors’ claim about obtaining LSE via a specific flag may be directionally correct but could be challenged if the reviewer expects exact API signatures or code-level evidence.
- “Q-only sparsity is more GPU-friendly than QK sparsity” needs harder evidence. They provide plausible architectural arguments (SIMT efficiency, irregular memory access, filling missing scores, ToCa being slower), but a reviewer might still want microbenchmarks/profiling or more kernel-level evidence to fully buy the claim.

**Reviewer Scores:**

Reviewer nMzk

Original rating: 6 (marginal accept)

Likely change: increase to 8 (accept) or remain 6

Reason: Their core concern (EA cost) is addressed with concrete reductions and amortization, plus evidence of portability and generalization. The distillation-combination question is also answered empirically.

Reviewer vjj7

Original rating: 6 (marginal accept)

Likely change: increase to 8 or remain 6

Reason: They fix the table typo and add SVG2 comparisons. The LSE justification is strengthened with both statistics and a theoretical bound, though it may still be seen as conditional.

Reviewer eRBx

Original rating: 4 (marginal reject)

Likely change: 4

Reason: Hyperparameter sensitivity is addressed cleanly, but the deeper concerns, “why EA” and “missed spatio-temporal structural optimizations,” are structural and not fully resolved by rebuttal. The integration with SVG helps framing, yet may not convince them the method choice is theoretically/strategically optimal.

Reviewer hZdk

Original rating: 4 (marginal reject)

Likely change: 4 or 6

Reason: They directly answer implementation questions (LSE access), clarify the “GPU-friendly” claim, provide a performance breakdown, and fix a contradiction. These actions typically reduce “uncertainty-driven” rejects. Remaining risk is if the reviewer is strict on kernel/API exactness and demands more concrete evidence.

---

### Decision · Program_Chairs · 2026-01-26

Accept (Poster)